# Class-wise Balancing Data Replay for Federated Class-Incremental Learning

**Zhuang Qi[1], Ying-Peng Tang[2], Lei Meng[1,*], Han Yu[2], Xiaoxiao Li[3,4], Xiangxu Meng[1]**
[1]School of Software, Shandong University, China
[2]College of Computing and Data Science, Nanyang Technological University, Singapore
[3]Department of Electrical and Computer Engineering, University of British Columbia, Canada
[4] Vector Institute, Canada
z_qi@mail.sdu.edu.cn, yingpeng.tang@ntu.edu.sg, lmeng@sdu.edu.cn,
han.yu@ntu.edu.sg, xiaoxiao.li@ece.ubc.ca, mxx@sdu.edu.cn

## Abstract

Federated Class Incremental Learning (FCIL) aims to collaboratively process continuously increasing incoming tasks across multiple clients. Among various approaches, data replay has become a promising solution, which can alleviate forgetting by reintroducing representative samples from previous tasks. However, their performance is typically limited by class imbalance, both within the replay buffer due to limited global awareness and between replayed and newly arrived classes. To address this issue, we propose a class-wise balancing data replay method for FCIL (`FedCBDR`), which employs a global coordination mechanism for class-level memory construction and reweights the learning objective to alleviate the aforementioned imbalances. Specifically, `FedCBDR` has two key components: 1) the global-perspective data replay module reconstructs global representations of prior task in a privacy-preserving manner, which then guides a class-aware and importance-sensitive sampling strategy to achieve balanced replay; 2) Subsequently, to handle class imbalance across tasks, the task-aware temperature scaling module adaptively adjusts the temperature of logits at both class and instance levels based on task dynamics, which reduces the model's overconfidence in majority classes while enhancing its sensitivity to minority classes. Experimental results verified that `FedCBDR` achieves balanced class-wise sampling under heterogeneous data distributions and improves generalization under task imbalance between earlier and recent tasks, yielding a 2%-15% Top-1 accuracy improvement over six state-of-the-art methods.

## 1 Introduction

Federated learning (FL) is a distributed machine learning paradigm that enables collaborative training of a shared global model across multiple data sources [1, 2, 3, 4, 5]. It periodically performs parameter-level interaction between clients and the server instead of gathering clients' data, which can enhance data privacy while leveraging the diversity of distributed data sources to build a more generalized global model [6, 7, 8, 9, 10, 11]. This mechanism makes it widely applicable to various fields [12, 13, 14, 15, 16]. Building upon this foundation, Federated Class-Incremental Learning (FCIL) extends FL by introducing dynamic data streams where clients sequentially encounter different task classes under non-independent and identically distributed data [17, 18, 19, 20, 21]. However, this amplifies the inherent complexities of FL, as the global model must integrate heterogeneous and

---

*Corresponding author

39th Conference on Neural Information Processing Systems (NeurIPS 2025).

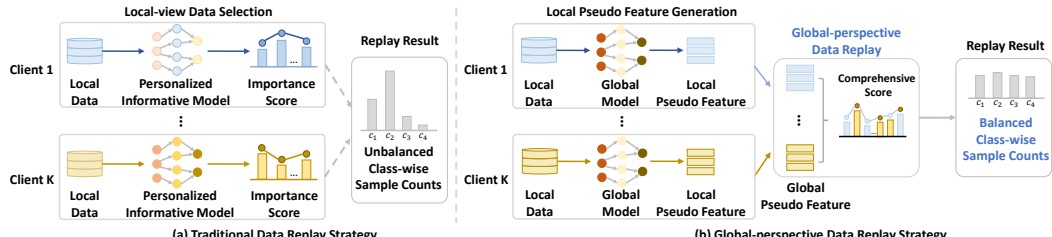

Figure 1: Motivation of the `FedCBDR`. Traditional data replay strategies typically focus on local information and, due to the lack of global awareness, often result in imbalanced class distributions during replay. `FedCBDR` aims to explore global information in a privacy-preserving manner and leverage it for sampling, which can alleviate the class imbalance problem.

evolving knowledge from clients while mitigating catastrophic forgetting, despite having no or only limited access to historical data [22, 23, 24].

To address the challenge of catastrophic forgetting in FCIL, data replay has emerged as a promising strategy for retaining knowledge from previous tasks. Existing replay-based methods can be broadly categorized into two types: generative-based replay and exemplar-based replay. The former leverages generative models to synthesize representative samples from historical tasks [23, 25, 26]. Its core idea is to learn the data distribution of previous tasks and internalize knowledge in the form of model parameters, enabling the indirect reconstruction of prior knowledge through sample generation when needed [19, 27]. However, they often overlook the computational cost of training generative models and are inherently constrained by the quality and fidelity of the synthesized data [23, 28, 29]. In contrast, exemplar-based replay methods directly store real samples from previous tasks, avoiding the complexity of generative processes while leveraging high-quality raw data to ensure robust retention of prior knowledge [17, 28, 30, 31]. These methods rely on a limited set of historical samples to maintain the decision boundaries of previously learned task classes. However, due to the lack of a global perspective on data distribution across clients, these methods are prone to class-level imbalance in replayed samples, which undermines the model's ability to retain prior knowledge [30, 31].

To address these issues, this paper proposes a class-wise balancing data replay method for FCIL, termed `FedCBDR`, which incorporates the global signal to regulate class-balanced memory construction, aiming to achieve distribution-aware replay and mitigate the challenges posed by non-IID client data, as illustrated in Figure 1. Specifically, `FedCBDR` comprises two primary modules: 1) the global-perspective data replay (GDR) module reconstructs a privacy-preserving pseudo global representation of historical tasks by leveraging feature space decomposition, which enables effective cross-client knowledge integration while preserving essential attributes information. Furthermore, it introduces a principled importance-driven selection mechanism that enables class-balanced replay, guided by a globally-informed understanding of data distribution; 2) the task-aware temperature scaling (TTS) module introduces a multi-level dynamic confidence calibration strategy that combines task-level temperature adjustment with instance-level weighting. By modulating the sharpness of the softmax distribution, it balances the predictive confidence between majority and minority classes, enhancing the model's robustness to class imbalance between historical and current task samples.

Extensive experiments were conducted on three datasets with different levels of heterogeneity, including performance comparisons, ablation studies, in-depth analysis, and case studies. The results demonstrate that `FedCBDR` effectively balances the number of replayed samples across classes and alleviates the long-tail problem. Compared to six state-of-the-art existing methods, `FedCBDR` achieves a 2%-15% Top-1 accuracy improvement.

## 2 Related Work

### 2.1 Exemplar-based Replay Methods

In FCIL, exemplar-based replay methods aim to mitigate catastrophic forgetting by storing and replaying a subset of samples from previous tasks. They typically maintain a small exemplar buffer on each client, which is used during training alongside new task data to preserve knowledge of

previously learned classes [17, 30, 31, 32, 33, 34]. For example, GLFC alleviates forgetting in FCIL by leveraging local exemplar buffers for rehearsal, while introducing class-aware gradient compensation and prototype-guided global coordination to jointly address local and global forgetting [17]. Moreover, Re-Fed introduces a Personalized Informative Model to strategically identify and replay task-relevant local samples, enhancing the efficiency of buffer usage and further reducing forgetting in heterogeneous client environments [30]. However, the lack of global insight in local sample selection often results in class imbalance, while the long-tailed distribution between replayed and current data is frequently overlooked, degrading the effectiveness of data replay [30, 31].

## 2.2 Generative-based Replay Methods

Generative replay methods aim to reconstruct the samples of past tasks through techniques such as generative modeling [19, 23, 27, 35, 36], which enables the model to revisit historical knowledge to mitigate catastrophic forgetting. Following this line of thought, TARGET generates pseudo features through a globally pre-trained encoder and performs knowledge distillation by aligning the current model's predictions with those of a frozen global model [27]; LANDER utilizes pre-trained semantic text embeddings as anchors to synthesize meaningful pseudo samples, and distills knowledge by aligning the model's predictions with class prototypes derived from textual descriptions [19]. However, these methods are typically limited by the high computational cost of training generative models and the suboptimal performance caused by low-fidelity pseudo samples [19, 27].

## 2.3 Knowledge Distillation-based Methods

Knowledge distillation-based methods generally follow two paradigms. The first focus om aligning the output predictions of the current model with those of previous models, which aims to preserve task-specific decision boundaries [37, 38, 39, 40, 41, 42, 43, 44, 45]. The second estimates the importance of model parameters for previously learned tasks and performs regularization to prevent forgetting [46, 47]. Both approaches avoid storing raw data but are prone to knowledge degradation over time, especially as the number of tasks increases [37, 46].

## 3 Preliminaries

We consider a federated class-incremental learning (FCIL) setting, where a central server aims to collaboratively train a global model with the assistance of $K$ distributed clients. Each client $k$ receives a sequence of classification tasks $\{\mathcal{D}_k^{(1)}, \mathcal{D}_k^{(2)}, \ldots, \mathcal{D}_k^{(t)}\}$, where each task introduces a disjoint set of new classes. Upon the arrival of task $t$, the global model parameters $\theta_t$ are optimized to minimize the average loss over the union of all samples seen so far, i.e., $\mathbb{D}^t = \bigcup_{s=1}^{t} \bigcup_{k=1}^{K} \mathcal{D}_k^{(s)}$, by solving $\min_\theta \frac{1}{|\mathbb{D}^t|} \sum_{s=1}^{t} \sum_{k=1}^{K} \sum_{i=1}^{N_k^{(s)}} \mathcal{L}(f_k(x_{k,i}^{(s)}; \theta), y_{k,i}^{(s)})$.

In replay-based methods, each client maintains a memory buffer with a fixed budget of $M$ samples. When task $t$ arrives, the client selects up to $N$ representative samples from each of the previous tasks $\{1, \ldots, t-1\}$, subject to the total memory constraint. The resulting memory set is denoted by $\mathcal{B}_k^{(t-1)} = \bigcup_{s=1}^{t-1} \{(x_{k,i}^{(s)}, y_{k,i}^{(s)})\}_{i=1}^{N}$, where $N$ is the number of samples stored per task and $\mathcal{B}_k^{(t-1)}$ satisfies $|\mathcal{B}_k^{(t-1)}| \leq M$. The local training set on client $k$ then becomes $\mathcal{D}_{k,\text{train}}^{(t)} = \mathcal{D}_k^{(t)} \cup \mathcal{B}_k^{(t-1)}$, combining current and replayed samples. Based on these local datasets, the server updates the global model by minimizing the aggregated loss: $\min_\theta \sum_{k=1}^{K} \sum_{(x,y) \in \mathcal{D}_{k,\text{train}}^{(t)}} \mathcal{L}(f_k(x; \theta), y)$.

## 4 Class-wise Balancing Data Replay for Federated Class-Incremental Learning

This section presents an effective active data selection method for FCIL, which aims to explore global data distribution to balance class-wise sampling. Moreover, it leverages temperature scaling to adjust the logits, which can alleviate the imbalance between samples from previously learned and newly introduced tasks. Figure 2 and Algorithm 1 illustrates the framework of the proposed FedCBDR.

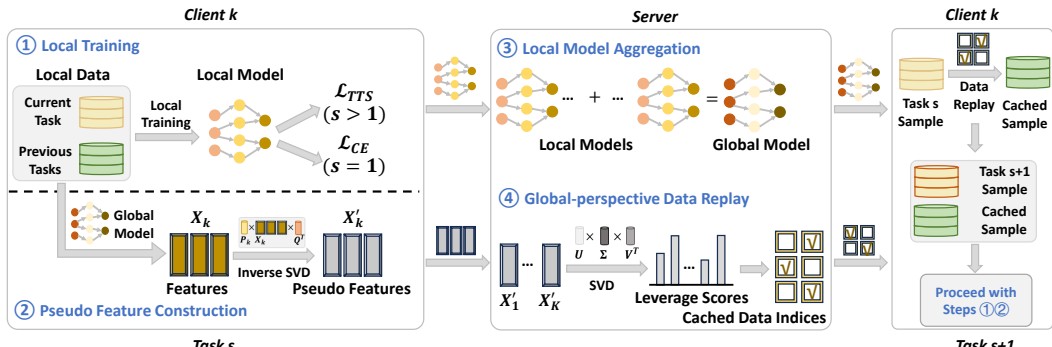

Figure 2: Illustration of the FedCBDR framework. It first trains local models using samples from current and previous tasks. After a fixed number of communication rounds, each client extracts local sample features using the global model and applies inverse singular value decomposition (ISVD) to obtain pseudo features. The server then aggregates both local models and pseudo features, performs SVD on the features, and selects representative samples based on leverage scores. The corresponding sample indices are sent back to the clients for balanced replay.

## 4.1 Global-perspective Data Replay (GDR)

Due to privacy constraints, traditional data replay strategies typically rely on local data distributions. However, the absence of global information often leads to class imbalance in the replay buffer. To address this, the GDR module aggregates local informative features into a global pseudo feature set, enabling exploration of the global distribution without exposing raw data.

Inspired by Singular Value Decomposition (SVD) [48, 49], we first generate a set of random orthogonal matrices: a client-specific matrix $P_k^{(i)} \in \mathbb{R}^{|\mathcal{D}_k^{(i)}| \times |\mathcal{D}_k^{(i)}|}$ for each client $k$ and task $i$, and a globally shared matrix $Q^{(i)} \in \mathbb{R}^{d \times d}$, where $d$ denotes the dimension of the feature. Each client encrypts its local feature matrix $X_k^{(i)} = M_g(\mathcal{D}_k^{(i)})$ via Inverse Singular Value Decomposition (ISVD):

$$X_k^{(i)'} = P_k^{(i)} X_k^{(i)} Q^{(i)}, \tag{1}$$

and uploads the encrypted matrix $X_k^{(i)'}$ to the server, where $M_g(\cdot)$ is the feature extractor of the global model. The server then aggregates all encrypted matrices into a global matrix $X^{(i)'}$:

$$X^{(i)'} = \text{concat}\{X_k^{(i)'} \mid k = 1, \dots, K\}, \tag{2}$$

and performs SVD as follows:

$$X^{(i)'} = U^{(i)'} \Sigma^{(i)'} V^{(i)'\top}, \tag{3}$$

where $U^{(i)'} \in \mathbb{R}^{n \times n}$, $\Sigma^{(i)'} \in \mathbb{R}^{n \times d}$, and $V^{(i)'} \in \mathbb{R}^{d \times d}$, with $n$ denoting the total number of samples from all clients. Next, the server extracts a submatrix of the left singular vectors for each client $k$ by:

$$U_k^{(i)} = \mathcal{I}_k(U^{(i)'}) \in \mathbb{R}^{|\mathcal{D}_k^{(i)}| \times n}, \tag{4}$$

where $\mathcal{I}_k(\cdot)$ denotes a row selection function that returns the indices corresponding to client $k$'s samples. To quantify the importance of local samples within the global latent space, client $k$ computes a leverage score [50, 51, 52] for $j$-th sample of task $i$ as:

$$\tau_k^{i,j} = \|e_{i,j}^\top U_k^{(i)}\|_2^2, \tag{5}$$

where $e_{i,j}$ denotes the $j$-th standard basis vector in task $i$. Notably, a higher leverage score indicates that the sample has a larger projection in the low-dimensional latent space, suggesting that it contributes more significantly to the global structure and is more representative. Moreover, clients send their leverage scores to the server, which aggregates them into a global vector $\tau^i = \text{concat}\{\tau_k^{i,j} | k = 1, ..., K; j = 1, ..., n_k^i\}$ and normalizes it to obtain a sampling distribution:

$$p_k^{i,j} = \frac{\tau_k^{i,j}}{\sum_{j'=1}^{n_k^i} \tau_k^{i,j'}}. \tag{6}$$

---

**Algorithm 1** FEDCBDR

---

1: **Initialize:** $R$: number of communication rounds; $K$: number of clients; $t$: number of tasks; $\theta_g$: global model parameters; $\mathcal{B}_k^{pre}$: replay buffer for historical tasks on client $k$; $\mathcal{D}_k^s$: local data of task $s$ on client $k$.
2: **for** each task $s = 1$ to $t$ **do**
3:    **for** each communication round $r = 1$ to $R$ **do**
4:       **for** each client $k = 1$ to $K$ **do**
5:          Initialize local model parameters: $\theta_k \leftarrow \theta_g$
6:          **if** $s == 1$ **then**
7:             Sample a mini-batch $\zeta$ from $\mathcal{D}_k^{(1)}$, and update $\theta_k$ using Eq. 8.
8:          **else**
9:             Store the historical task data corresponding to globally sampled IDs into $\mathcal{B}_k^{pre}$.
10:             Sample a mini-batch $\zeta$ from $\mathcal{D}_k^{(s)} \cup \mathcal{B}_k^{pre}$, and update $\theta_k$ using Eq. 9.
11:             Compute pseudo-features based on Eq. 1, and upload them to the server.
12:          **end if**
13:       **end for**
14:       **if** $r < R$ **then**
15:          Aggregate local model parameters across clients.
16:       **else**
17:          Aggregate model parameters and pseudo-features from all clients using Eq. 2.
18:          Perform **Global Sampling** based on Eqs. 3–6, and send the selected sample IDs back to the corresponding clients.
19:       **end if**
20:    **end for**
21: **end for**
22: **// Global Sampling Procedure**
23: Form the global feature pool $X^{(i)}$ by aggregating all pseudo-features via Eq. (2).
24: Perform singular value decomposition (SVD) using Eq. 3 to extract key attributes.
25: Compute leverage scores for each client's samples using Eqs. 4–5, and normalize globally using Eq. 6.
26: Perform sampling and adjust the probabilities of the selected samples accordingly.

---

Subsequently, we perform i.i.d. sampling based on the distribution $\mathbf{p} = \{p_k^{i,j} | k = 1, ..., K; j = 1, ..., n_k^i\}$. Once a sample $x$ is selected, its sampling weight is adjusted to $\frac{1}{\sqrt{n_s \cdot p_x}} e_x$, where $n_s$ denotes the number of selected samples and $p_x$ is the original sampling probability of $x$, $e_x$ is the standard basis vector of $x$. This adjustment ensures unbiased estimation during aggregation. Following the sampling procedure, the server communicates the selected sample indices to their respective clients, where the corresponding data points are subsequently marked for further use.

## 4.2 Task-aware Temperature Scaling (TTS)

Due to limited replay budgets, samples from previous tasks are often much fewer than those from the current task, leading to class imbalance and poor retention of past knowledge. To mitigate this, the TTS module dynamically adjusts sample temperature and weight based on task order, enhancing the contribution of tail-class samples during optimization.

Specifically, we use a lower temperature to sharpen logits for samples from earlier tasks. Furthermore, to further amplify the optimization effect of tail-class samples during training, we also leverage a re-weighted cross-entropy loss, i.e.,

$$\mathcal{L}_{\text{TTS}} = \frac{1}{N_{\text{old}}} \sum_{i=1}^{N_{\text{old}}} \omega_{\text{old}} \cdot \text{CE}\left(y_i, \text{Softmax}\left(\text{Concat}\left(\frac{z_i^{\text{old}}}{\tau_{\text{old}}}, \frac{z_i^{\text{new}}}{\tau_{\text{new}}}\right)\right)\right) + \frac{1}{N_{\text{new}}} \sum_{j=1}^{N_{\text{new}}} \omega_{\text{new}} \cdot \text{CE}\left(y_j, \text{Softmax}\left(\text{Concat}\left(\frac{z_j^{\text{old}}}{\tau_{\text{old}}}, \frac{z_j^{\text{new}}}{\tau_{\text{new}}}\right)\right)\right) \quad (7)$$

where $N_{\text{old}}$ and $N_{\text{new}}$ denote the number of samples from the previous and newly arrived task, respectively; $y_i$ and $y_j$ are the ground-truth labels; $z_i^{\text{old}}$ and $z_i^{\text{new}}$ denote the logits corresponding to old classes and new classes, respectively; $\tau_{\text{old}}$ and $\tau_{\text{new}}$ are the temperature scaling factors for previous and newly arrived task samples; $\omega_{\text{old}}$ and $\omega_{\text{new}}$ are the corresponding sample weights; $\text{CE}(\cdot)$ denotes

Table 1: Statistics of the datasets used in experiments.

| Datasets | #Class | #Training | #Testing | Image Size | Federated Settings | | |
|---|---|---|---|---|---|---|---|
| | | | | | Clients | Tasks | Heterogeneity |
| CIFAR10 | 10 | 50,000 | 10,000 | $32 \times 32$ | 5/10 | 3/5 | 0.5/1.0 |
| CIFAR100 | 100 | 50,000 | 10,000 | $32 \times 32$ | 5/10 | 5/10 | 0.1/0.5/1.0 |
| TinyImageNet | 200 | 100,000 | 10,000 | $64 \times 64$ | 5/10 | 10/20 | 0.1/0.5/1.0 |

the cross-entropy loss function; and $\mathrm{Softmax}(z/\tau)$ is the temperature-scaled softmax function used to adjust the sharpness of the output distribution.

### 4.3 Training Strategy

The training strategy consists of two stages to progressively address the evolving challenges in federated class-incremental learning. Algorithm 1 presents the pipeline of the FedCBDR.

**Stage 1: Initial Task Optimization.** In the first task, client $k$ learns from local data using the standard cross-entropy loss, i.e.,

$$\min_{\theta_k} \frac{1}{N} \sum_{i=1}^{N} \mathrm{CE}(y_i, \mathrm{Softmax}(f_{\theta_k}(x_i))), \tag{8}$$

**Stage 2: Class-Incremental Optimization.** As new tasks arrive and class imbalance emerges between previous and current tasks in client $k$, we employ $\mathcal{L}_{TTS}$ to mitigate the imbalance, i.e.,

$$\min_{\theta_k} \frac{1}{N_{\mathrm{old}}} \sum_{i=1}^{N_{\mathrm{old}}} \omega_{\mathrm{old}} \cdot \mathrm{CE}\left(y_i, \mathrm{Softmax}\left(\mathrm{Concat}\left(\frac{f_{\theta_k}^{\mathrm{old}}(x_i)}{\tau_{\mathrm{old}}}, \frac{f_{\theta_k}^{\mathrm{new}}(x_i)}{\tau_{\mathrm{new}}}\right)\right)\right) + \frac{1}{N_{\mathrm{new}}} \sum_{j=1}^{N_{\mathrm{new}}} \omega_{\mathrm{new}} \cdot \mathrm{CE}\left(y_j, \mathrm{Softmax}\left(\mathrm{Concat}\left(\frac{f_{\theta_k}^{\mathrm{old}}(x_j)}{\tau_{\mathrm{old}}}, \frac{f_{\theta_k}^{\mathrm{new}}(x_j)}{\tau_{\mathrm{new}}}\right)\right)\right) \tag{9}$$

where $x_i$ is the input sample, $y_i$ is the corresponding ground-truth, $f_{\theta_k}^{\mathrm{old}}(x)$ and $f_{\theta_k}^{\mathrm{new}}(x)$ represent the outputs of the model corresponding to old and new classes, respectively. $\mathrm{Softmax}(\cdot)$ converts the logits into a probability distribution.

## 5 Experiments

### 5.1 Experiment Settings

**Datasets.** Following existing studies [27, 30], we conducted all experiments on three commonly used datasets, including CIFAR10 [53, 54], CIFAR100 [53, 54] and TinyImageNet [55] to validate the effectiveness of the FedCBDR. We simulate heterogeneous data distributions across clients using the Dirichlet distribution with parameters $\beta = \{0.1, 0.5, 1.0\}$, where smaller values of $\beta$ correspond to higher level of data heterogeneity. The statistical details are presented in the Table 1.

**Evaluation Metric.** Following prior studies [19, 56, 57, 58], we adopt Top-1 Accuracy as the evaluation metric, defined as $\mathrm{Accuracy} = N_{\mathrm{correct}}/N_{\mathrm{total}}$, where $N_{\mathrm{correct}}$ and $N_{\mathrm{total}}$ denote the number of correct predictions and the total number of samples, respectively.

**Implementation Details.** In the experiments, the number of clients is fixed at $K = 5$, with each client running local epochs $E = 2$ per round, using a batch size $B = 128$. For all datasets, we adopt ResNet-18 as the backbone, with the classifier's output dimension dynamically updated as tasks progress and conduct $T = 100$ communication rounds per task. The SGD optimizer is employed with a learning rate of $0.01$ and a weight decay of $1 \times 10^{-5}$. The number of stored samples per task varies by dataset and split setting: for CIFAR10, 450 samples are stored under 3-task splits and 300 under 5-task splits; for CIFAR100, 1,000 samples are used for 5-task splits and 500 for 10-task splits; for TinyImageNet, 2,000 samples are stored for 10-task splits and 1,000 for 20-task splits. For the temperature and weighted parameters, we select $\tau_{old} \in \{0.8, 0.9\}$ and $w_{old} \in \{1.1, 1.2, 1.3, 1.4\}$ for previous tasks, while $\tau_{new} \in \{1.1, 1.2\}$ and $w_{new} \in \{0.7, 0.8, 0.9\}$ are used for newly arrived tasks. Moreover, the hyperparameters of baselines are tuned based on their original papers for fair comparison. And training on each client is performed using an NVIDIA RTX 3090 GPU (24 GB).

Table 2: Performance comparison between `FedCBDR` and baselines across three datasets under varying levels of heterogeneity ($\beta$). CIFAR10 is divided into 3 tasks, CIFAR100 into 5 tasks, and TinyImageNet into 10 tasks. All methods were executed under three different random seeds, and both the mean and standard deviation of the results are reported. The best results are **bolded**.

| Method | CIFAR10 | | CIFAR100 | | | TinyImageNet | | |
|---|---|---|---|---|---|---|---|---|
| | $\beta$=0.5 | $\beta$=1.0 | $\beta$=0.1 | $\beta$=0.5 | $\beta$=1.0 | $\beta$=0.1 | $\beta$=0.5 | $\beta$=1.0 |
| Finetune | $38.71_{\pm3.7}$ | $40.49_{\pm3.0}$ | $15.17_{\pm2.2}$ | $16.75_{\pm2.6}$ | $17.15_{\pm1.3}$ | $6.06_{\pm0.9}$ | $6.00_{\pm0.8}$ | $6.40_{\pm0.5}$ |
| FedEWC | $39.93_{\pm1.1}$ | $42.70_{\pm2.5}$ | $18.30_{\pm2.4}$ | $20.70_{\pm5.3}$ | $21.22_{\pm3.4}$ | $6.30_{\pm0.8}$ | $6.94_{\pm0.7}$ | $7.36_{\pm0.6}$ |
| FedLwF | $56.03_{\pm1.6}$ | $58.29_{\pm3.6}$ | $33.97_{\pm2.6}$ | $37.09_{\pm3.1}$ | $41.91_{\pm2.5}$ | $11.81_{\pm0.9}$ | $11.47_{\pm1.0}$ | $14.87_{\pm1.2}$ |
| TARGET | $44.17_{\pm4.4}$ | $54.49_{\pm4.5}$ | $30.15_{\pm3.6}$ | $33.47_{\pm4.3}$ | $35.25_{\pm2.0}$ | $10.71_{\pm1.4}$ | $10.18_{\pm0.9}$ | $12.49_{\pm1.1}$ |
| LANDER | $53.90_{\pm3.2}$ | $60.79_{\pm1.4}$ | $44.07_{\pm3.3}$ | $47.63_{\pm3.7}$ | $\mathbf{52.77}_{\pm1.4}$ | $13.80_{\pm0.8}$ | $15.02_{\pm1.9}$ | $16.36_{\pm1.0}$ |
| Re-Fed | $53.46_{\pm3.5}$ | $60.73_{\pm4.3}$ | $32.67_{\pm3.7}$ | $38.42_{\pm2.9}$ | $45.28_{\pm2.6}$ | $15.73_{\pm1.7}$ | $15.93_{\pm1.3}$ | $16.05_{\pm1.1}$ |
| FedCBDR | $\mathbf{64.11}_{\pm1.2}$ | $\mathbf{65.20}_{\pm1.9}$ | $\mathbf{46.40}_{\pm1.6}$ | $\mathbf{49.76}_{\pm2.7}$ | $52.06_{\pm1.5}$ | $\mathbf{18.37}_{\pm1.1}$ | $\mathbf{18.86}_{\pm0.9}$ | $\mathbf{18.78}_{\pm0.9}$ |

Table 3: Performance comparison between `FedCBDR` and baselines across three datasets under varying levels of heterogeneity ($\beta$). CIFAR10 is divided into 5 tasks, CIFAR100 into 10 tasks, and TinyImageNet into 20 tasks. All methods were executed under three different random seeds, and both the mean and standard deviation of the results are reported. The best results are **bolded**.

| Method | CIFAR10 | | CIFAR100 | | | TinyImageNet | | |
|---|---|---|---|---|---|---|---|---|
| | $\beta$=0.5 | $\beta$=1.0 | $\beta$=0.1 | $\beta$=0.5 | $\beta$=1.0 | $\beta$=0.1 | $\beta$=0.5 | $\beta$=1.0 |
| Finetune | $19.78_{\pm2.3}$ | $23.34_{\pm2.8}$ | $7.22_{\pm1.1}$ | $9.39_{\pm0.7}$ | $9.64_{\pm0.5}$ | $3.40_{\pm0.4}$ | $3.73_{\pm0.5}$ | $3.95_{\pm0.3}$ |
| FedEWC | $20.11_{\pm2.7}$ | $28.97_{\pm2.3}$ | $8.08_{\pm0.3}$ | $11.69_{\pm0.7}$ | $12.19_{\pm1.7}$ | $3.50_{\pm0.3}$ | $4.58_{\pm0.4}$ | $5.08_{\pm0.9}$ |
| FedLwF | $38.76_{\pm2.3}$ | $52.95_{\pm3.1}$ | $18.73_{\pm1.1}$ | $25.30_{\pm0.6}$ | $28.21_{\pm1.0}$ | $3.67_{\pm0.4}$ | $6.61_{\pm0.6}$ | $10.22_{\pm1.3}$ |
| TARGET | $35.27_{\pm1.7}$ | $48.28_{\pm1.2}$ | $13.61_{\pm0.8}$ | $21.09_{\pm0.4}$ | $24.22_{\pm1.1}$ | $5.32_{\pm0.6}$ | $5.39_{\pm0.6}$ | $5.72_{\pm0.5}$ |
| LANDER | $40.22_{\pm2.4}$ | $58.07_{\pm3.4}$ | $27.79_{\pm1.9}$ | $33.51_{\pm2.3}$ | $37.42_{\pm1.8}$ | $8.89_{\pm0.6}$ | $8.57_{\pm0.8}$ | $10.45_{\pm0.6}$ |
| Re-Fed | $54.94_{\pm3.1}$ | $58.19_{\pm2.5}$ | $29.33_{\pm1.3}$ | $39.54_{\pm1.3}$ | $40.96_{\pm1.1}$ | $9.36_{\pm0.9}$ | $11.44_{\pm0.7}$ | $12.27_{\pm1.1}$ |
| FedCBDR | $\mathbf{61.18}_{\pm1.3}$ | $\mathbf{65.42}_{\pm1.8}$ | $\mathbf{45.11}_{\pm1.2}$ | $\mathbf{46.51}_{\pm1.6}$ | $\mathbf{47.79}_{\pm1.4}$ | $\mathbf{12.58}_{\pm0.4}$ | $\mathbf{14.47}_{\pm0.7}$ | $\mathbf{15.69}_{\pm0.6}$ |

## 5.2 Performance Comparison

To evaluate the effectiveness of the proposed `FedCBDR`, we compare it with six representative baseline methods: Finetune [19], FedEWC [46], FedLwF [37], TARGET [27], LANDER [19], and Re-Fed [30]. As reported in Table 2 and Table 3, the results can be summarized as follows:

- `FedCBDR` achieves the highest Top-1 accuracy in most cases across the three datasets under varying levels of heterogeneity and task splits. The only suboptimal result occurs on CIFAR100 with 5 tasks and $\beta = 1.0$, where `FedCBDR` (52.06%) performs slightly worse than LANDER (52.77%). This demonstrates the adaptability and robustness of the proposed `FedCBDR` across complex settings.

- Despite LANDER attains the best performance on CIFAR100 under the 5-task and $\beta = 1.0$ setting, it demands the generation of more than 10,000 samples per task, and the overhead of training its data generator surpasses that of the federated model, raising concerns about its scalability.

- Knowledge distillation-based methods like FedLwF perform well on simpler tasks (CIFAR10) by using pretrained knowledge to guide local models. However, their performance drops on more complex or heterogeneous tasks due to limited adaptability to local variations.

- Given an equal memory budget, class-balanced sampling (`FedCBDR`) consistently achieves superior performance compared to class-imbalanced strategy (Re-Fed), as it ensures more equitable representation across categories and effectively mitigates class-level forgetting in FCIL scenarios.

## 5.3 Ablation Study

In this section, we conducted an ablation study to investigate the contributions of key modules, including the Global-perspective Active Data Replay (GDR) module and the Task-aware Temperature Scaling (TTS) module. Table 4 presents the results, which can be summarized as follows:

Table 4: Ablation results under different levels of data heterogeneity and task splitting settings. "3/5/10" denotes CIFAR10 with 3 tasks, CIFAR100 with 5 tasks, and TinyImageNet with 10 tasks; "5/10/20" represents 5, 10, and 20 tasks respectively.

| Task Splitting | Method | CIFAR10 | | CIFAR100 | | | TinyImageNet | | |
|---|---|---|---|---|---|---|---|---|---|
| | | $\beta$=0.5 | $\beta$=1.0 | $\beta$=0.1 | $\beta$=0.5 | $\beta$=1.0 | $\beta$=0.1 | $\beta$=0.5 | $\beta$=1.0 |
| 3/5/10 | Finetune | $38.71_{\pm3.7}$ | $40.49_{\pm3.0}$ | $15.17_{\pm2.2}$ | $16.75_{\pm2.6}$ | $17.15_{\pm1.3}$ | $6.06_{\pm0.9}$ | $6.00_{\pm0.8}$ | $6.40_{\pm0.5}$ |
| | +GDR | $62.13_{\pm2.1}$ | $63.81_{\pm1.9}$ | $45.28_{\pm1.5}$ | $47.66_{\pm0.9}$ | $51.47_{\pm1.7}$ | $17.24_{\pm0.6}$ | $17.89_{\pm0.5}$ | $18.04_{\pm0.4}$ |
| | +TTS | $41.34_{\pm2.3}$ | $42.55_{\pm2.2}$ | $17.32_{\pm0.5}$ | $17.14_{\pm0.4}$ | $19.32_{\pm0.5}$ | $6.67_{\pm0.2}$ | $6.92_{\pm0.3}$ | $7.27_{\pm0.4}$ |
| | +GDR+TTS | $\mathbf{64.11}_{\pm1.2}$ | $\mathbf{65.20}_{\pm1.9}$ | $\mathbf{46.40}_{\pm1.6}$ | $\mathbf{49.76}_{\pm2.7}$ | $\mathbf{52.06}_{\pm1.5}$ | $\mathbf{18.37}_{\pm1.1}$ | $\mathbf{18.86}_{\pm0.9}$ | $\mathbf{18.78}_{\pm0.9}$ |
| 5/10/20 | Finetune | $19.78_{\pm2.3}$ | $23.34_{\pm2.8}$ | $7.22_{\pm1.1}$ | $9.39_{\pm0.7}$ | $9.64_{\pm0.5}$ | $3.40_{\pm0.4}$ | $3.73_{\pm0.5}$ | $3.95_{\pm0.3}$ |
| | +GDR | $59.34_{\pm3.1}$ | $63.20_{\pm2.6}$ | $44.04_{\pm1.3}$ | $46.33_{\pm0.5}$ | $46.50_{\pm0.8}$ | $11.44_{\pm0.3}$ | $13.85_{\pm0.5}$ | $14.51_{\pm0.6}$ |
| | +TTS | $22.43_{\pm2.4}$ | $25.81_{\pm2.1}$ | $8.31_{\pm0.2}$ | $10.21_{\pm0.3}$ | $10.33_{\pm0.4}$ | $3.78_{\pm0.5}$ | $4.04_{\pm0.4}$ | $4.16_{\pm0.3}$ |
| | +GDR+TTS | $\mathbf{61.18}_{\pm1.3}$ | $\mathbf{65.42}_{\pm1.8}$ | $\mathbf{45.11}_{\pm1.2}$ | $\mathbf{46.51}_{\pm1.6}$ | $\mathbf{47.79}_{\pm1.4}$ | $\mathbf{12.58}_{\pm0.4}$ | $\mathbf{14.47}_{\pm0.7}$ | $\mathbf{15.69}_{\pm0.6}$ |

- Incorporating the GDR module substantially improves performance across all cases, particularly under high data heterogeneity ($\beta = 0.1$), demonstrating its effectiveness in alleviating catastrophic forgetting even with a limited number of replay samples in federated class-incremental learning.

- Using the TTS module alone leads to consistent improvements over Finetune, highlighting its effectiveness in addressing intra-client class imbalance through temperature scaling. This contribution to better generalization is particularly evident under the more challenging "5/10/20" task splitting scenario.

- The integration of both modules results in the best overall performance, consistently achieving the highest Top-1 accuracy across various datasets and heterogeneity levels. This stems from their complementary strengths: the GDR module mitigates inter-task forgetting, while the TTS module alleviates both intra- and inter-client class imbalance.

## 5.4 Performance Evaluation of `FedCBDR` under Incremental Tasks

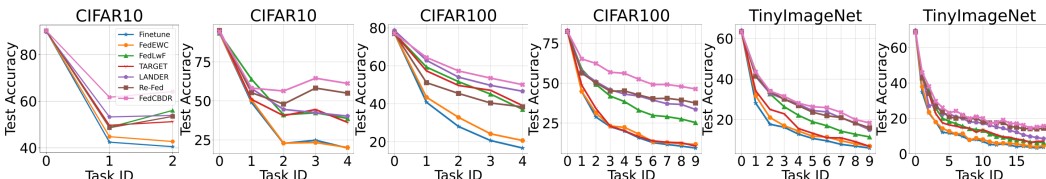

Figure 3: Performance comparison of all methods across varying task splits on CIFAR10 (3/5 tasks), CIFAR100 (5/10 tasks), and TinyImageNet (10/20 tasks) with $\beta = 0.5$.

This section investigates the performance of `FedCBDR` and the baselines in incremental cases on three datasets. Figure 3 presents the average accuracy of all methods on both current and previous tasks. Notably, `FedCBDR` consistently outperforms other baseline methods across all task splits, with its accuracy curves remaining higher throughout the incremental process. Furthermore, `FedCBDR` exhibits a slower performance degradation as the number of tasks increases, indicating stronger resistance to catastrophic forgetting. In addition, it maintains significantly higher accuracy on later tasks, especially in challenging settings such as CIFAR100 and TinyImageNet with 10 tasks, highlighting its ability to balance knowledge retention and adaptation to new classes.

## 5.5 Quantitative Analysis of Replay Buffer Size on Test Accuracy

In this section, we evaluate the performance of Re-Fed and `FedCBDR` under different buffer size $M$ settings, and additionally include LANDER, which generates 10,240 synthetic samples for each task. As shown in Table 5, `FedCBDR` exhibits more significant performance advantages over Re-Fed under limited memory settings, and even surpasses LANDER, which relies on a large-scale generative replay buffer. Furthermore, as the buffer size increases, `FedCBDR` demonstrates more stable and significant performance improvements. This indicates that the method can effectively leverage larger replay buffers for continuous optimization. However, Re-Fed exhibits noticeable performance fluctuations

Table 5: Comparison of model performance with varying memory size $M$ across datasets.

| Methods | CIFAR10 | | | CIFAR100 | | | TinyImageNet | |
|---------|---------|---------|---------|----------|----------|----------|--------------|--------|
| | M=150 | M=300 | M=450 | M=500 | M=1000 | M=1500 | M=2000 | M=2500 |
| LANDER (10240) | | 52.90 | | | 47.05 | | | 14.77 | |
| Re-Fed | 47.23 | 53.47 | 54.66 | 33.89 | 38.42 | 47.84 | 15.89 | 16.78 |
| FedCBDR | 51.99 | 59.02 | 63.81 | 40.12 | 49.66 | 55.94 | 18.33 | 19.41 |

under small and medium buffer settings. In particular, its accuracy is significantly lower than that of FedCBDR on CIFAR100 with $M = 500$ and TinyImageNet with $M = 2000$, indicating its limited ability to mitigate inter-class interference and retain knowledge from previous tasks. These findings validate that, under the same buffer budget, a balanced sampling distribution is more effective than an imbalanced one in alleviating forgetting and improving overall model performance.

## 5.6 Sensitivity Analysis of FedCBDR on Temperature and Weighted Hyperparameters

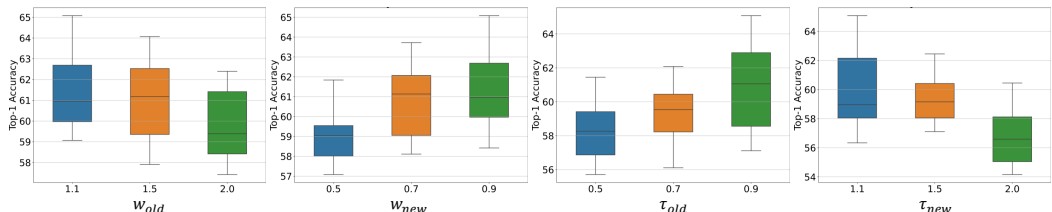

Figure 4: Performance of FedCBDR on CIFAR100 ($\beta = 0.5$, 5-task split) under varying temperature ($\tau_{old} \in \{0.5, 0.7, 0.9\}$, $\tau_{new} \in \{1.1, 1.5, 2.0\}$) and weighted ($w_{old} \in \{1.1, 1.5, 2.0\}$, $w_{new} \in \{0.5, 0.7, 0.9\}$) settings.

Figure 4 gives a sensitivity analysis of FedCBDR with respect to temperature and sample weighting hyperparameters. Overall, temperature scaling and sample re-weighting help mitigate class imbalance, but model performance varies considerably with different hyperparameter settings. The model achieves better overall performance when $\omega_{old} = 1.1$, $\omega_{new} = 0.9$, $\tau_{old} = 0.9$, and $\tau_{new} = 1.1$. This is because slightly higher weight and temperature for previous-task samples help retain old knowledge, while lower weight and higher temperature for newly arrived samples reduce overfitting and improve adaptation. However, inappropriate hyperparameter choices may harm performance. For instance, a large $\tau_{new}$ (e.g., 2.0) leads to overly smooth predictions, reducing discrimination among newly arrived classes. These results emphasize the need for proper tuning to ensure balanced learning.

## 5.7 Comparison of Per-Class Sample Distributions in the Replay Buffer

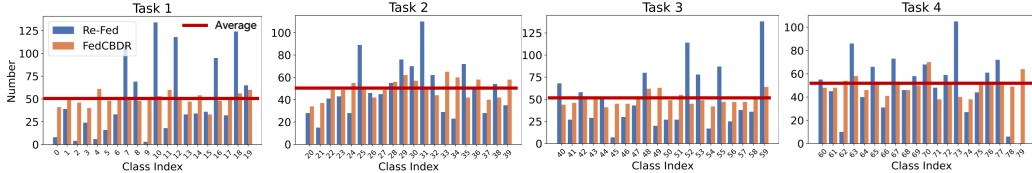

Figure 5: Comparison of per-class sample distributions in the replay buffer between FedCBDR and Re-Fed on the CIFAR100 dataset, under a heterogeneity level of $\beta = 0.5$ and a 5-task split case.

To evaluate the effectiveness of FedCBDR in balancing class-wise sampling, Figure 5 illustrates the per-class sample distributions in the replay buffer between FedCBDR and Re-Fed across different task stages. Overall, across different task stages, FedCBDR (orange bars) exhibits a per-class sample distribution that is consistently closer to the average level (red line), whereas Re-Fed shows noticeable skewness and fluctuations. This indicates that FedCBDR is more effective in achieving balanced class-wise sampling in the replay buffer. In addition, FedCBDR ensures that no class is overlooked

during sampling, while Re-Fed may fail to retain certain classes in the replay buffer—for example, class 79 is missing in Task 4 under Re-Fed. This highlights the robustness of FedCBDR in maintaining class coverage throughout incremental learning.

## 5.8 Visualization of Model Attention and Temperature-aware Logits Adjustment

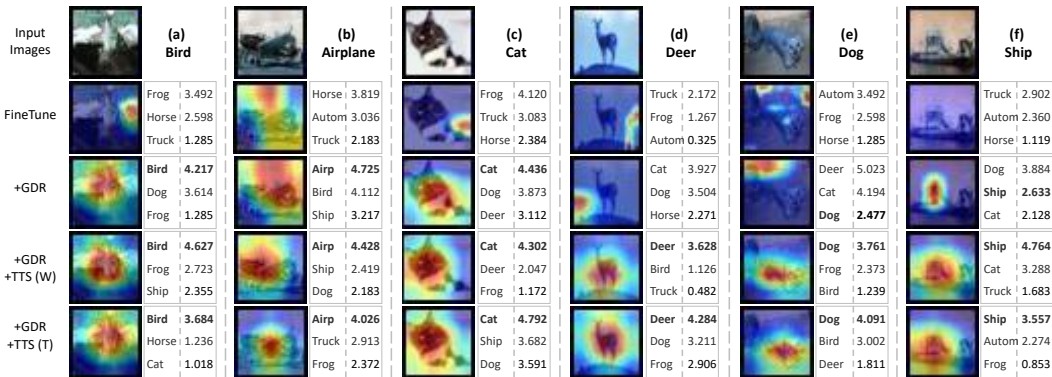

Figure 6: Case studies of model attention and the effect of temperature-aware logits adjustment on CIFAR10 ($\beta = 0.5$, 3-task split).

This section presents case studies comparing prediction confidence and attention focus using Grad-CAM [59, 60, 61, 62, 63] visualizations. As shown in Figure 6(a-c), in the absence of data replay, the model struggles to correctly classify samples from previous tasks and fails to attend to the relevant target regions. The incorporation of data replay in FedCBDR alleviates this issue by correcting predictions and guiding attention back to semantically important areas. Despite partially mitigating forgetting, data replay alone may still lead to misclassification or low-confidence predictions for tail classes with limited samples. The integration of temperature scaling (T) and sample re-weighting (W) in the TTS module enables the model to better distinguish confusing classes through temperature adjustment, improving tail class accuracy and enhancing prediction stability, as depicted in Figure 6(d-f). These findings demonstrate the crucial role of the collaboration between both modules in mitigating knowledge forgetting during incremental learning.

## 6 Conclusions and Future Work

To address the challenge of inter-class imbalance in replay-based federated class-incremental learning, we propose FedCBDR that combines class-balanced sampling with loss adjustment to better exploit the global data distribution and enhance the contribution of tail-class samples to model optimization. Specifically, it uses SVD to decouple and reconstruct local data, aggregates local information in a privacy-preserving manner, and explores i.i.d. sampling within the aggregated distribution. In addition, it applies task-aware temperature scaling and sample re-weighting to mitigate the long-tail problem. Experimental results show that FedCBDR effectively reduces inter-class sampling imbalance and significantly improves final performance.

Despite the impressive performance of FedCBDR, there remain several directions worth exploring to address its limitations. Specifically, we plan to investigate lightweight sampling strategies to reduce feature transmission costs in FedCBDR, and to develop more robust post-sampling balancing methods that mitigate class imbalance with less sensitivity to hyperparameters [64, 65, 66, 67]. Moreover, extending FedCBDR to more complex scenarios [68, 69, 70, 71, 72] is a promising direction.

## Acknowledgments

This work is supported in part by the Key Research and Development Program of Shandong Province (Grant No. 2024TSGC0667) and the Ministry of Education, Singapore, under its Academic Research Fund Tier 1 (RG101/24).

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

# A Appendix

## A.1 Experimental Results

### A.1.1 Performance Comparison

To thoroughly verify the effectiveness of the proposed `FedCBDR`, we compare its performance against various baselines under the setting of 10 clients. Based on the original implementations, we generate 10,240 synthetic samples per task for both TARGET and LANDER. The data replay configurations for Re-Fed and `FedCBDR` follow the settings outlined in Section 5.1. The results are presented in Tables 6 and 7. Consistent with the results shown in Tables 2 and 3, `FedCBDR` achieves the best performance across all cases. Notably, `FedCBDR` **achieves over a 10% gain compared to the second-best performing method in several settings.**

Table 6: Performance comparison between `FedCBDR` and baseline methods across CIFAR-10, CIFAR-100, and TinyImageNet under varying levels of data heterogeneity (Dirichlet parameter $\beta$). Specifically, CIFAR-10 is split into 3 tasks, CIFAR-100 into 5 tasks, and TinyImageNet into 10 tasks. The number of clients is fixed at 10, and all experiments are conducted with a random seed of 2023 to ensure reproducibility. The best results are **bolded**.

| Method | CIFAR10 | | CIFAR100 | | | TinyImageNet | | |
|---|---|---|---|---|---|---|---|---|
| | $\beta$=0.5 | $\beta$=1.0 | $\beta$=0.1 | $\beta$=0.5 | $\beta$=1.0 | $\beta$=0.1 | $\beta$=0.5 | $\beta$=1.0 |
| FedEWC | 36.40 | 42.00 | 15.19 | 18.66 | 19.50 | 6.19 | 7.23 | 7.78 |
| FedLwF | 48.24 | 49.11 | 27.02 | 37.92 | 41.77 | 10.67 | 13.02 | 14.73 |
| TARGET | 38.23 | 41.11 | 18.34 | 23.59 | 25.71 | 7.45 | 8.29 | 8.87 |
| LANDER | 41.54 | 45.52 | 30.83 | 43.69 | 47.29 | 12.33 | 15.18 | 15.64 |
| Re-Fed | 45.49 | 52.22 | 31.81 | 36.40 | 37.95 | 9.28 | 11.48 | 12.10 |
| `FedCBDR` | 59.80 | 62.59 | 42.25 | 47.90 | 48.55 | 14.81 | 16.54 | 17.43 |

Table 7: Performance comparison between `FedCBDR` and baseline methods across CIFAR-10, CIFAR-100, and TinyImageNet under varying levels of data heterogeneity (Dirichlet parameter $\beta$). Specifically, CIFAR-10 is split into 5 tasks, CIFAR-100 into 10 tasks, and TinyImageNet into 20 tasks. The number of clients is fixed at 10, and all experiments are conducted with a random seed of 2023 to ensure reproducibility. The best results are **bolded**.

| Method | CIFAR10 | | CIFAR100 | | | TinyImageNet | | |
|---|---|---|---|---|---|---|---|---|
| | $\beta$=0.5 | $\beta$=1.0 | $\beta$=0.1 | $\beta$=0.5 | $\beta$=1 | $\beta$=0.1 | $\beta$=0.5 | $\beta$=1.0 |
| FedEWC | 20.18 | 23.33 | 6.68 | 10.98 | 12.30 | 3.27 | 4.80 | 4.89 |
| FedLwF | 43.31 | 46.79 | 13.82 | 17.79 | 27.80 | 4.50 | 5.71 | 9.07 |
| TARGET | 21.60 | 28.39 | 12.11 | 16.64 | 17.14 | 3.45 | 4.88 | 5.01 |
| LANDER | 27.24 | 32.21 | 10.74 | 25.87 | 31.79 | 4.74 | 12.05 | 13.21 |
| Re-Fed | 38.28 | 39.22 | 28.08 | 33.52 | 37.27 | 7.95 | 8.53 | 10.13 |
| `FedCBDR` | 51.71 | 59.57 | 37.42 | 43.82 | 45.50 | 11.51 | 14.45 | 15.25 |

### A.1.2 Performance Evaluation of `FedCBDR` under Incremental Tasks

We evaluate the performance evolution of `FedCBDR` and competing methods under a 10-client setting across incremental tasks on three benchmark datasets. Specifically, CIFAR-10 is split into 3 tasks ($\beta = \{0.5, 1.0\}$), CIFAR-100 into 5 tasks ($\beta = \{0.1, 0.5, 1.0\}$), and TinyImageNet into 10 tasks ($\beta = \{0.1, 0.5, 1.0\}$). As shown in Figure 7, `FedCBDR` **consistently outperforms all baseline methods across incremental tasks, maintaining higher accuracy on both current and previous tasks throughout the training process**. Moreover, its performance degrades more slowly as the number of tasks increases.

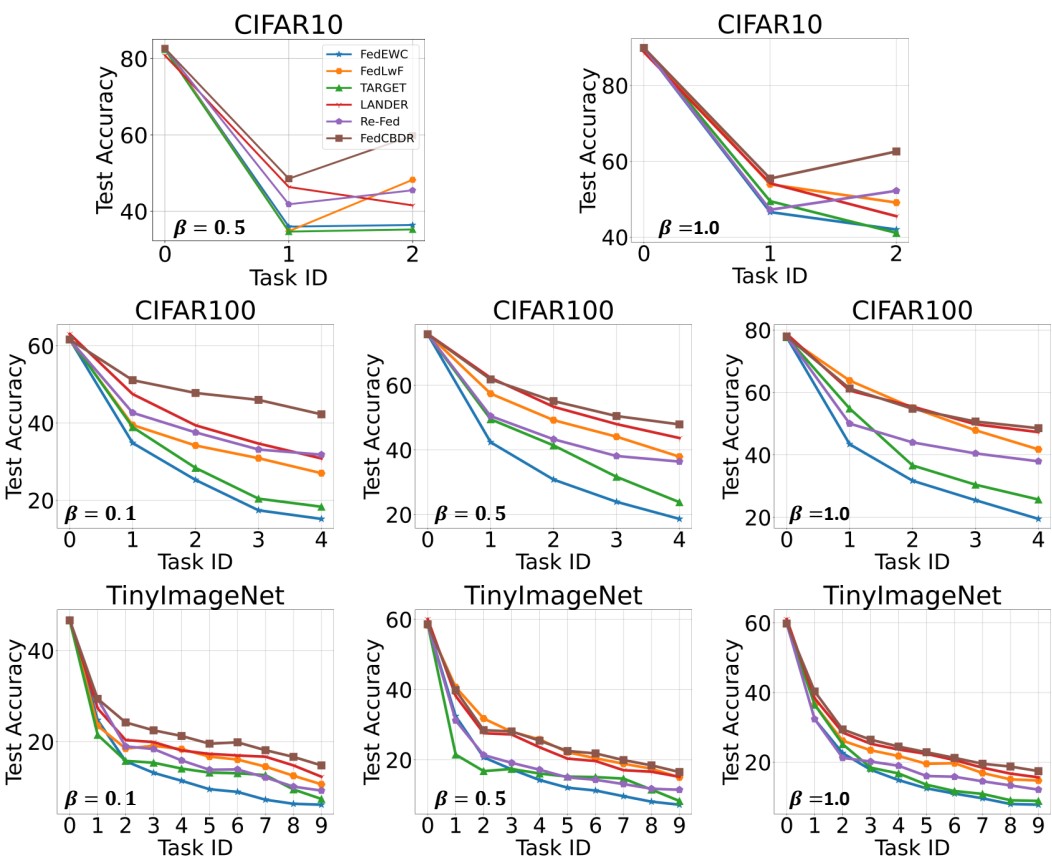

Figure 7: Comparison of per-class sample distributions in the replay buffer between `FedCBDR` and Re-Fed on the CIFAR100 dataset, conducted under a heterogeneity level of $\beta = 0.5$, with a 10-task split and 5 clients.

### A.1.3 Comparison of Per-Class Sample Distributions in the Replay Buffer

We further validate the capability of the proposed `FedCBDR` to balance per-class sample distributions in more complex scenarios. Specifically, we divide the CIFAR100 dataset into 10 tasks. As illustrated in Figure 8, Re-Fed exhibits substantial disparities in the number of replayed samples across classes. For example, in task 1, while classes 10 and 18 contain nearly 100 samples each, class 11 has fewer than 10. In contrast, `FedCBDR` **effectively alleviates such class imbalance, with the number of replayed samples for all classes remaining consistently close to the average (as marked by the red line)**. This contributes to more stable knowledge retention across tasks and enhances overall model generalization.

### A.1.4 Quantitative Analysis of Replay Buffer Size on Test Accuracy

Table 8: Comparison of model performance with varying replay budget $M$ per task across datasets, with the number of clients fixed at 10, and heterogeneity level $\beta = 0.5$.

| Methods | CIFAR10 | | | CIFAR100 | | |
|---|---|---|---|---|---|---|
| | M=150 | M=300 | M=450 | M=500 | M=1000 | M=1500 |
| Re-Fed | 39.22 | 42.93 | 45.49 | 28.21 | 36.40 | 41.78 |
| FedCBDR | 48.15 | 54.62 | 59.80 | 38.34 | 47.90 | 51.14 |

We compare the performance of the data replay-based methods, Re-Fed and `FedCBDR`, under varying replay buffer budgets. Specifically, for CIFAR10, the buffer size is adjusted among $\{150, 300, 450\}$, while for CIFAR100, it ranges from $\{500, 1000, 1500\}$. The number of clients is set to 10, and heterogeneity level $\beta = 0.5$. As shown in Table 8, **the performance of both methods improves as**

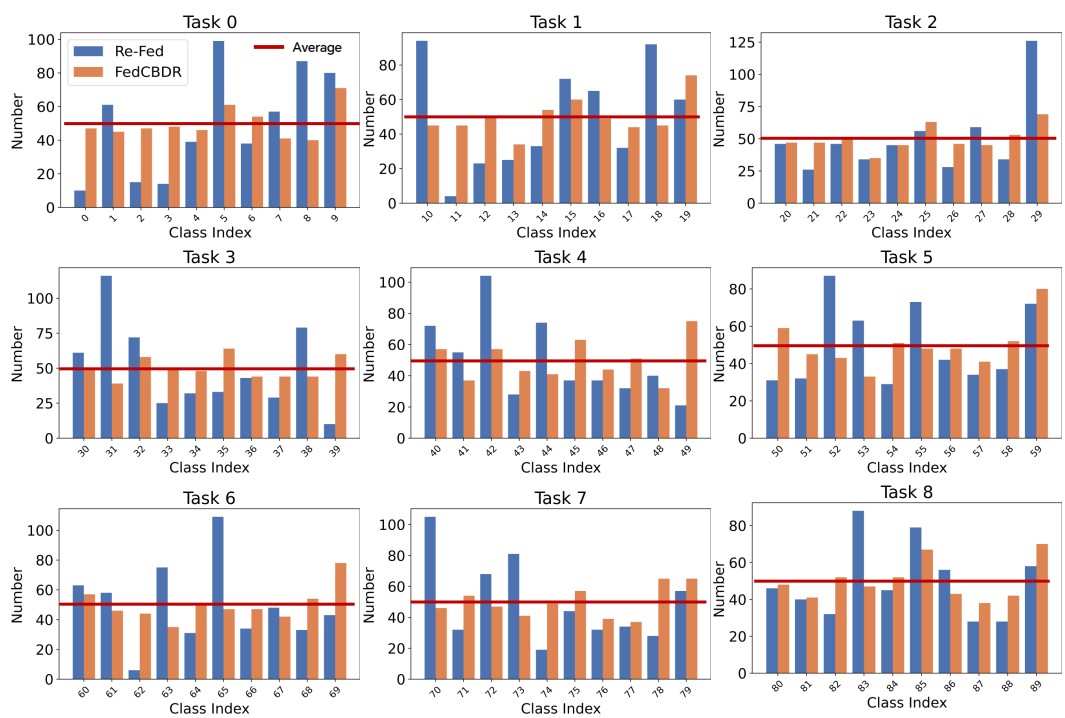

Figure 8: Comparison of per-class sample distributions in the replay buffer between `FedCBDR` and Re-Fed on the CIFAR100 dataset, conducted under a heterogeneity level of $\beta = 0.5$, with a 10-task split and 5 clients.

**the buffer size increases, with `FedCBDR` maintaining a clear advantage over Re-Fed under all settings.** This also underscores the importance of balancing per-class sample counts in the replay buffer to ensure fair representation and stable performance.

### A.1.5 Evaluation on the Impact of Local Training Epochs

To assess the impact of local training intensity, we compare the performance of LANDER, Re-Fed, and `FedCBDR`, under varying local training epoch settings. Specifically, the evaluation is conducted on CIFAR10 divided into 3 tasks and CIFAR100 divided into 5 tasks, under a federated setting with 10 clients and a heterogeneity level of $\beta = 0.5$. As shown in Figure 9, **both GDR and GDR+TTS consistently outperform the baseline methods (LANDER and Re-Fed) across all local training epoch settings on both CIFAR10 and CIFAR100**. Moreover, **GDR+TTS achieves the highest test accuracy in every configuration**. The improvement brought by TTS highlights its necessity in alleviating class imbalance during local training. And, unlike other methods whose performance drops at 10 local epochs due to biased updates, **GDR+TTS demonstrates a sustained improvement potential.**

### A.1.6 Performance Assessment of the Final Model Across Tasks

This section compares the final model performance of different methods (LANDER, Re-Fed, `FedCBDR`) across various tasks. Specifically, all experiments are conducted under a federated setting with 5 clients and a heterogeneity level of $\beta = 0.5$. CIFAR10 is split into 3 tasks and CIFAR100 into 5 tasks. Each task is trained for 50 communication rounds, with each client performing 2 local training epochs per round using a batch size of 128. For sample replay, LANDER synthesizes 10,240 samples per task, while Re-Fed and `FedCBDR` retain 150 and 1,000 real samples per task on CIFAR10 and CIFAR100, respectively. As shown in Table 9, **LANDER suffers from significant forgetting of earlier tasks**, as evidenced by its low accuracy of only 1.37% on Task 1 of CIFAR10. This indicates a severe inability to retain prior knowledge. Moreover, **LANDER also shows a noticeable decline in performance on the last task**, achieving only 57.00% on Task 5 of CIFAR100, suggesting that its generalization to new tasks is also limited under non-i.i.d. conditions. Compared to LANDER and

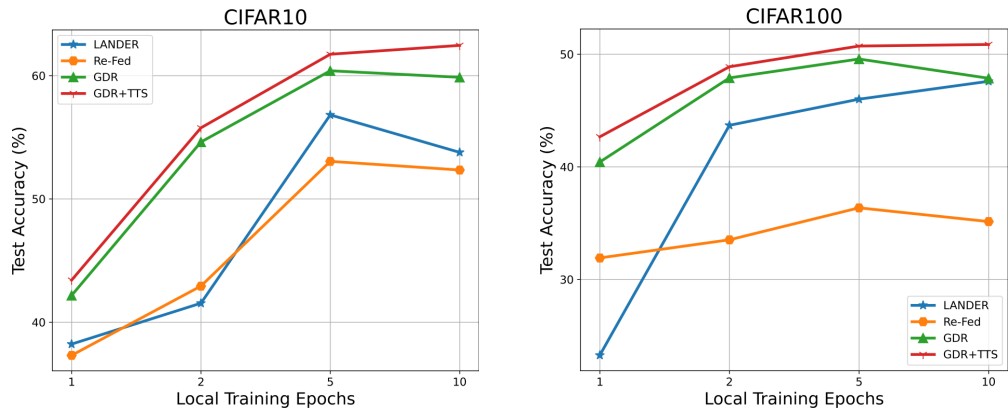

Figure 9: Comparison of the final performance of the LANDER, Re-Fed, GDR, and GDR+TTS methods under different numbers of local training epochs. GDR and TTS are the two functional modules proposed in this work, and GDR+TTS=`FedCBDR`.

Re-Fed, **GDR significantly enhances the retention of knowledge from most early tasks**. **This demonstrates the advantage of balanced sample replay over imbalanced sampling**. In particular, **GDR+TTS outperforms GDR alone, highlighting the effectiveness of the proposed TTS module in mitigating class imbalance** and supporting long-term knowledge preservation under non-i.i.d. settings.

Table 9: Per-task and average accuracy (%) of different methods on CIFAR10 and CIFAR100.

|  | CIFAR10 | | | | CIFAR100 | | | | | |
|---|---|---|---|---|---|---|---|---|---|---|
|  | Task 1 | Task 2 | Task 3 | ALL | Task 1 | Task 2 | Task 3 | Task 4 | Task 5 | ALL |
| LANDER | 1.37 | 30.00 | 88.32 | 44.74 | 33.95 | 40.90 | 43.70 | 44.45 | 57.00 | 44.00 |
| Re-Fed | 14.20 | 18.33 | 95.88 | 44.28 | 23.40 | 22.00 | 21.10 | 34.10 | 81.70 | 36.46 |
| GDR | 17.07 | 19.00 | 96.10 | 49.26 | 40.40 | 37.90 | 39.25 | 47.50 | 80.45 | 49.10 |
| GDR+TTS | 21.43 | 21.46 | 96.08 | 51.30 | 41.45 | 38.05 | 38.50 | 48.55 | 81.40 | 49.59 |

### A.1.7   Scalability, Complexity, and Communication Efficiency

We have conducted additional large-scale experiments by extending the number of clients to 50/100 while simulating asynchronous participation, and the client sampling rate is set to 0.2. The dataset is divided into five tasks, and in Re-Fed and FedCBDR, the number of replay samples is set to 150/300 for CIFAR-10 and 500/1000 for CIFAR-100.The following results can be summarized:

Table 10: Comparison of continual federated learning methods with 50 clients on CIFAR10 and CIFAR100 under different Dirichlet data partitions ($\alpha$).

| **50 Clients** | CIFAR10 | | CIFAR100 | |
|---|---|---|---|---|
|  | $\alpha = 0.5$ | $\alpha = 1.0$ | $\alpha = 0.5$ | $\alpha = 1.0$ |
| FedLwF | 17.51 | 19.43 | 19.99 | 23.91 |
| LANDER | 18.08 | 21.27 | 26.96 | 27.34 |
| Re-Fed (300/500) | 29.65 | 32.45 | 27.12 | 28.86 |
| FedCBDR (300/500) | 34.76 | 35.98 | 28.75 | 30.90 |
| Re-Fed (600/1000) | 36.40 | 38.46 | 35.94 | 37.79 |
| FedCBDR (600/1000) | **41.33** | **45.31** | **38.54** | **39.69** |

Table 11: Computation and communication analysis on CIFAR10 and CIFAR100 datasets. Top: computation overhead per client; Bottom: communication cost per round.

| Dataset | #Clients (K) | Samples/ Client ($n_k$) | Feature Dim ($d$) | ISVD Time/Client (s) | Feature Extraction Time/Client (s) | Total Extra Time/Client (s) |
|---------|--------------|-------------------------|-------------------|----------------------|------------------------------------|------------------------------|
| CIFAR10 | 5 | 2000 | 512 | 0.00580 | 0.9083 | 0.9141 |
| CIFAR100 | 5 | 2000 | 512 | 0.00452 | 1.0250 | 1.2952 |

| Dataset | #Clients (K) | Samples/ Client ($n_k$) | Feature Dim ($d$) | Upload/Client (MB) | Total Upload per Round (MB) | Index Download (KB) |
|---------|--------------|-------------------------|-------------------|--------------------|-----------------------------|---------------------|
| CIFAR10 | 10 | 100 | 512 | 3.906 | 19.53 | 39.06 |
| CIFAR10 | 10 | 100 | 128 | 0.9765 | 4.8825 | 39.06 |
| CIFAR10 | 10 | 100 | 32 | 0.2441 | 1.2206 | 39.06 |

Moreover, we have further clarified the computational complexity of the SVD step. Client-side ISVD costs $\mathcal{O}(n_k d^2)$ per client, while server-side SVD costs $\mathcal{O}(N d^2)$, where $N = \sum_k n_k$ is the total number of pseudo-features.

We also provide a summary of the additional training cost on clients, the communication overhead, and the server-side computation incurred by our method in comparison to FedAvg. **Overall, the additional cost of performing SVD and computing leverage scores is relatively small.**

