# OpenReview forum: "Class-wise Balancing Data Replay for Federated Class-Incremental Learning"
_NeurIPS.cc/2025/Conference — NeurIPS 2025 oral_

### Official Review · Reviewer_i4C7 · 2025-06-18

**Clarity:** 2
**Significance:** 2
**Originality:** 2
**Rating:** 4
**Confidence:** 4

**Summary:**

This paper proposes FedCBDR, a new method to address catastrophic forgetting in federated class-incremental learning (FCIL) by explicitly tackling class imbalance issues. The authors identify two sources of imbalance in replay-based FCIL: (i) imbalance within the replay buffer due to clients selecting examples without global awareness, and (ii) imbalance between replayed old classes and newly introduced classes in incremental tasks. To mitigate these, FedCBDR introduces two key components. First, a global-perspective data replay (GDR) module constructs a pseudo global representation of prior tasks in a privacy-preserving way and uses it to guide class-balanced exemplar selection across clients. Second, a task-aware temperature scaling (TTS) module dynamically calibrates prediction logits (with task-specific temperature adjustment and instance-level weighting) to reduce overconfidence in majority classes while boosting minority classes. The method allows each client to incorporate a globally informed, class-balanced replay selection and applies confidence scaling to handle long-tail class distributions over time. Experimental results show that FedCBDR yields significantly improved retention of old classes and overall accuracy compared to existing methods. Evaluations on three benchmark image classification datasets (CIFAR-10, CIFAR-100, Tiny-ImageNet) under various non-i.i.d. settings demonstrate 2%–15% higher Top-1 accuracy than six state-of-the-art FCIL approaches.

**Questions:**

#1. Pseudo-Feature Construction: The mechanism for constructing and using pseudo global features in the GDR module could be described more clearly. Currently, it's mentioned that each client performs an inverse SVD (ISVD) on local features to produce pseudo-features, and then the server aggregates and applies SVD to select representative samples. This is a complex procedure – for instance, what exactly are the "pseudo-features" mathematically, and how is inverse SVD used? It would help if the paper included a concise algorithm or a toy example illustrating this process step-by-step. In particular, formally and rigorously analyzing how privacy is preserved (e.g. are the random orthogonal bases used in ISVD shared or kept secret? Can the server reconstruct original data from the pseudo-features?) is essential.

#2. Comparison to GLFC or Related Methods: As noted in the weaknesses, GLFC is a closely related prior work tackling a similar problem. We invite the authors to clarify how FedCBDR differs from and improves upon GLFC (and any other prior FCIL methods with global coordination). Is the key difference that FedCBDR achieves true class-balanced replay, whereas GLFC focuses on gradient alignment? If so, emphasizing this distinction in the introduction or related work would sharpen the paper's contribution. Adding a direct experimental comparison would be necessary.

#3. TTS Hyperparameter Sensitivity & Selection: The Task-aware Temperature Scaling (TTS) module's performance is shown to be sensitive to the choice of its hyperparameters: τ_old,τ_new,w_old,w_new (as indicated in Figure 4 ). (i) The paper provides specific tuning ranges used in the experiments (e.g., τ_old∈{0.8,0.9}). How were these particular ranges determined? Was it based on prior work, preliminary experiments, or other heuristics? (ii) Is there a more principled approach or a robust heuristic for setting these hyperparameters that goes beyond dataset-specific grid search? For instance, could these parameters be adapted based on observable task statistics, such as class imbalance ratios or model confidence levels on old versus new tasks?

#4. Computational Cost & Scalability of GDR: The GDR module involves client-side ISVD and server-side SVD operations, as well as communication of features and scores. What is the theoretical computational complexity of these operations? Furthermore, how much data (in terms of volume, e.g., bytes) needs to be communicated per round or per task between the clients and the server (e.g., for pseudo-features, leverage scores, selected sample indices)? Crucially, how do these computational and communication costs scale with key parameters such as the number of clients (K), the number of samples per client, the dimensionality of the features (d), and the total number of tasks? We invite the authors to provide a theoretical complexity analysis for the main computational steps in GDR. Additionally, providing empirical measurements (e.g., wall-clock time for computations, actual data volume transmitted) from the experimental setup would be highly beneficial.

#5. Realistic Federated Settings: The authors are encouraged to discuss how FedCBDR might extend to more realistic FL scenarios. For example, what if the number of clients is very large or if clients drop in and out (asynchronous participation)? Does the method require any synchronization (e.g. all clients must send features at the end of each task) that could be problematic if some clients are slow or offline?

**Ethical Concerns:**

["NO or VERY MINOR ethics concerns only"]

**Final Justification:**

The authors answered most of my questions through two rounds of rebuttal. I am happy to raise my score to 4. Borderline Accept.

**Limitations:**

**Author-Acknowledged Limitations:**

#1. Feature Transmission Costs in FedCBDR: The authors state an intention to "investigate lightweight sampling strategies to reduce feature transmission costs in FedCBDR". This implicitly acknowledges that the current GDR module, which involves clients sending transformed features $X_k^{(i)'}$ to the server, may incur significant communication overhead. To better motivate this future work and provide context, it would be beneficial to quantify the current feature transmission costs.

#2. Hyperparameter Sensitivity of Balancing Methods (TTS): The plan to "develop more robust post-sampling balancing methods that mitigate class imbalance with less sensitivity to hyperparameters"  directly addresses the observation that the TTS module's effectiveness can depend on careful tuning (Figure 4).

#3. Sampling for Globally Imbalanced Distributions: The authors identify that "designing sampling strategies for globally imbalanced distributions remains an open problem". It would be helpful to clarify how this "globally imbalanced distribution" problem differs from the class imbalance issues that FedCBDR currently aims to tackle (i.e., imbalance within replay buffers and between replayed and new task samples). If this refers to a more extreme or persistent form of imbalance that affects all tasks and clients from the outset, it would be useful to frame the scope of the current work more precisely against this larger, acknowledged challenge.

**Missing or Under-Discussed Limitations:**

#1. Depth of Privacy Guarantees for GDR: The assertion that GDR is "privacy-preserving" is not sufficiently substantiated with rigorous analysis. This stands as a critical limitation for a method proposed for federated learning, where privacy is a cornerstone. The authors should undertake a formal privacy analysis of the GDR module (e.g., within the differential privacy framework or by analyzing resistance to specific attack models).

#2. Server-Side Computational Bottleneck for GDR: The SVD operation performed on the aggregated pseudo-feature matrix $X_k^{(i)'}$ on the server can become a computational bottleneck, particularly as the number of participating clients, the number of samples per client, or the dimensionality of the features increases. This potential limitation should be explicitly discussed. The authors could suggest potential mitigation strategies for future work, such as the use of approximate SVD methods, incremental SVD, or randomized SVD techniques that might offer a better trade-off between accuracy and computational cost. Defining the scale (in terms of number of clients, total samples, feature dimensions) at which FedCBDR is currently practical based on their experimental setup would also be informative.

#3. Generality of Datasets: The experiments are focused on image classification benchmarks with relatively small images. It remains to be seen if the approach would scale to more complex data (e.g. high-resolution images like ImageNet-1k). We invite the authors to evaluate FedCBDR on ImageNet-1k dataset.

#4. Assumption of Synchronous Communication: The training pipeline described in Algorithm 1 implies a synchronous federated learning setup, where the server waits for updates (local models and pseudo-features) from all participating clients before proceeding with aggregation and the SVD operation. Real-world federated learning systems often have to contend with asynchronous client participation, device dropouts, or stragglers. We invite the authors to discuss whether the FedCBDR framework can be adapted to asynchronous settings or acknowledge the reliance on synchronous communication as a current limitation.

**Paper Formatting Concerns:**

No concerns

**Quality:**

2

**Strengths And Weaknesses:**

**Quality Strengths:**

The technical approach of FedCBDR appears valid. The two primary components, GDR and TTS, are motivated in the context of the class imbalance problem prevalent in FCIL. While the application of SVD for identifying globally salient features and the use of temperature scaling for calibrating the learning objective is well-established machine learning techniques, they are in this paper combined and applied to the specific challenges of the FCIL setting.

**Quality Weaknesses:**

#1. The claim that the GDR module is "privacy-preserving" is not formally substantiated with a rigorous privacy analysis. While raw data samples are not directly shared with the server, the ISVD process involves a client-specific matrix $P_k^{(i)}$ and a globally shared matrix $Q^{(i)}$ in the transformation $X_k^{(i)'}=P_k^{(i)} X_k^{(i)} Q^{(i)}$. The subsequent aggregation of these transformed features $X_k^{(i)'}$ at the server, followed by SVD, might still be vulnerable to various inference attacks or information leakage about the original client data. The paper cites references in the context of SVD, which pertain to "efficient decentralized federated singular vector decomposition" and "practical lossless federated singular vector decomposition" respectively. However, the specific privacy properties of these methods and a clear explanation of how they apply to the FedCBDR framework to ensure privacy are not adequately detailed. We invite the authors to present a formal and rigorous data privacy analysis (e.g., differential privacy guarantees).

#2. For the experiments, the absence of GLFC (Global-Local Forgetting Compensation, NeurIPS 2023) or its contemporary equivalents is critical, though they are discussed in related work. Given that GLFC specifically addresses forgetting in FCIL with a form of global coordination, it would be a natural baseline to include. We invite the authors to present a comparison with it. Similarly, one might wonder about simple baselines like FedAvg + replay buffer with uniform sampling (without FedCBDR's balancing) as a point of reference – although the paper does compare to Re-Fed, which is a strong replay-based method, an even simpler baseline could show the raw effect of class imbalance.

**Clarity Strengths:**

In general, the paper is well-written and structured. The problem formulation and related work give a clear background on FCIL and existing replay strategies. The method is illustrated with a framework diagram (Figure 2) and detailed step-by-step description, which aids understanding of the GDR and TTS modules.

**Clarity Weaknesses:**

#1. The description of the SVD/ISVD process within the GDR module could be more precise. Labeling the transformation $X_k^{(i)'}=P_k^{(i)} X_k^{(i)} Q^{(i)}$ as an "encryption" might be misleading if not used in a formal cryptographic sense. Further clarification on this transformation (e.g., feature obfuscation or rotation for privacy) and more details on the generation, properties (e.g., orthogonality), and secure sharing mechanism for the matrices $P_k^{(i)}$ and $Q^{(i)}$ are necessary.

#2. In addition, we invite the authors to clarify the dimensions of the matrix $\sum^{(i)'}$ resulting from the SVD operation.

#3. Besides, there appears to be a mismatch or mislabeling concerning Figure 7 and Figure 8 in the appendix. The captions for these two figures appear to be identical, while the figures themselves depict different types of information (performance curves versus sample distributions).

**Significance Strengths:**

The paper addresses a challenge in federated class-incremental learning – the class imbalance in replay data – which is often overlooked in prior work. By articulating the issue (both local replay bias and long-tailed new vs. old class distribution), the authors establish a motivation for FedCBDR. The proposed solution directly targets this gap.

**Significance Weaknesses:**

#1. Scalability concerns may limit the perceived significance of large-scale, real-world FL deployments. The experiments are conducted with a relatively small number of clients ($K=5$ in the main paper, $K=10$ in the appendix). The performance, computational feasibility (especially the server-side SVD in GDR), and communication efficiency of FedCBDR on much larger federated networks, potentially involving hundreds or thousands of clients, are not demonstrated.

#2. The sensitivity of the TTS module to its hyperparameters ($\tau_{old}, \tau_{new}, w_{old}, w_{new}$) is a practical concern. Figure 4 illustrates that performance can vary considerably with different hyperparameter settings, and inappropriate choices can be detrimental. If extensive and dataset-specific tuning is required for these hyperparameters in each new scenario, it reduces the method's ease of use and practical significance. Indeed, the authors also acknowledge this by proposing future work to develop methods with "less sensitivity to hyperparameters".

**Originality Strengths:**

The idea of incorporating a global perspective into replay buffer construction seems to be new in the FCIL context. Instead of each client selecting exemplars myopically, FedCBDR's GDR module uses a server-side aggregation of pseudo-features (via SVD-based feature decomposition) to guide class-balanced sampling across clients. This mechanism appears to be a novel combination of techniques (singular value decomposition for representative selection, leverage score sampling, etc.) not seen in prior FCIL methods. Similarly, the TTS module introduces a multi-level logit calibration scheme for federated learning, which seems to be an original way to handle class imbalance between tasks.

**Originality Weaknesses:**

#1. The method builds upon known techniques – exemplar replay and temperature scaling – by combining them with a global feature aggregation step. Some elements have precedents: for example, prior work GLFC (referenced as [9]) also introduced global coordination (via gradient compensation and prototype sharing) to address forgetting.

#2. FedCBDR's global pseudo-feature approach is different, but the authors could better clarify how their method fundamentally advances beyond such earlier approaches.

#3. The temperature scaling idea for class imbalance, while effective, is a relatively straightforward adaptation of calibration methods to this setting.

---

> ### Author Rebuttal · Authors · 2025-07-31
>
> We sincerely appreciate the professional comments provided by Reviewer i4C7
>
> W1: Privacy analysis
>
> A1: For any feature vector $x \in \mathbb{R}^d$, a random orthogonal matrix $P \in \mathbb{R}^{d \times d}$ satisfies:
> $$
> P^T P = P P^T = I.
> $$
> Hence, the orthogonal transformation $x' = P x$ preserves the L2-norm of the vector:
> $$
> \|x'\|_2 = \|P x\|_2 = \sqrt{x^{T} P^{T} P x} = \|x\|_2.
> $$
> This implies that all points on the unit sphere are merely rotated by $P$, with their original coordinate meaning destroyed. $P$ is sampled uniformly from the Haar distribution on $O(d)$, then $y = P x$ is effectively the representation of $x$ under an **unknown random rotation basis**.
> Without knowledge of $P$, an attacker observing $y$ cannot deduce the unique $x$, because:
> $$
> \forall x' \text{ with } \|x'\|_2 = \|x\|_2, \ \exists P' \in O(d), \ y = P' x'.
> $$
> In other words, infinitely many $x'$ on the sphere could lead to the same $y$ with an appropriate rotation $P'$.
>
> W2: Experiments with GLFC and the simple replay baseline
>
> A2: We conducted 5-task class-incremental experiments under different data heterogeneity settings and 5 clients. As shown in the Table, FedCBDR achieves the best performance across all cases.
>
> | Method           | CIFAR10 (α=0.5) | CIFAR10 (α=1.0) | CIFAR100 (α=0.5) | CIFAR100 (α=1.0) |
> |------------------|-----------------|-----------------|------------------|------------------|
> | **GLFC**         | 50.34 ± 2.8     | 60.21 ± 3.1     | 36.72 ± 2.5      | 44.69 ± 1.8      |
> | **FedAvg+Uniform** | 46.49 ± 4.2     | 55.57 ± 3.5     | 33.41 ± 3.6      | 44.73 ± 2.9      |
> | **Re-Fed**       | 53.46 ± 3.5     | 60.73 ± 4.3     | 38.42 ± 2.9      | 45.28 ± 2.6      |
> | **FedCBDR**      | **64.11 ± 1.2** | **65.20 ± 1.9** | **49.76 ± 2.7**  | **52.06 ± 1.5**  |
>
> W3: SVD/ISVD Transformation and Privacy
>
> A3: We replaced the term “encryption” with “orthogonal obfuscation,” and clearly defined the transformation:
>    $$
>    X_k^{(i)'} = P_k^{(i)} X_k^{(i)} Q^{(i)},
>    $$
>    where $P_k^{(i)}$ and $Q^{(i)}$ are orthogonal matrices preserving norms but rotating the feature space. And $P_k^{(i)}$ is generated locally on each client (via QR decomposition of random Gaussian matrices) and never transmitted to the server. The global $Q^{(i)}$ is initialized by the server and publicly shared but does not leak sensitive information.
>
> W4: Dimensions of $\Sigma^{(i)'}$
>
> A4: In the original manuscript, we wrote the full SVD as:
> $$
> X^{(i)'} = U^{(i)'} \Sigma^{(i)'} V^{(i)'T},
> $$
> with $U^{(i)'} \in \mathbb{R}^{n \times n}$, $\Sigma^{(i)'} \in \mathbb{R}^{n \times d}$,
> and $V^{(i)'} \in \mathbb{R}^{d \times d}$. While this is correct for the full SVD, in practice we only need the reduced SVD form with
> $$
> U^{(i)'} \in \mathbb{R}^{n \times r}, \
> \Sigma^{(i)'} \in \mathbb{R}^{r \times r}, \
> V^{(i)'} \in \mathbb{R}^{d \times r},
> $$
> where $r = \min(n, d)$. We will revise Section 4.1 to clarify this notation and explicitly define $r$ as the effective rank used for pseudo-feature construction.
>
> W5: Figure captions
>
> A5: We have revised the caption of Figure 7 to: “Performance comparison of all methods across varying settings on CIFAR10 (3 tasks, $\beta=\{0.1,0.5\}$), CIFAR100 (5 tasks, $\beta=\{0.1,0.5,1.0\}$), and TinyImageNet (10 tasks, $\beta=\{0.1,0.5,1.0\}$).”
>
> W6: Scalability, Complexity, and Communication Efficiency concerns
>
> A6: We have conducted additional large-scale experiments by extending the number of clients to 50/100 while simulating asynchronous participation, and the client sampling rate is set to 0.2. The dataset is divided into five tasks, and in Re-Fed and FedCBDR, the number of replay samples is set to 150/300 for CIFAR-10 and 500/1000 for CIFAR-100.The following results can be summarized:
> | 50 Clients            | CIFAR10 (alpha=0.5) | CIFAR10 (alpha=1.0) | CIFAR100 (alpha=0.5) | CIFAR100 (alpha=1.0) |
> |-----------------------|---------------------|----------------------|-----------------------|-----------------------|
> | FedLwf                | 17.51               | 19.43                | 19.99                 | 23.91                 |
> | LANDER                | 18.08               | 21.27                | 26.96                 | 27.34                 |
> | Re-Fed (300/500)      | 29.65               | 32.45                | 27.12                 | 28.86                 |
> | FedCBDR (300/500)     | 34.76               | 35.98                | 28.75                 | 30.90                 |
> | Re-Fed (600/1000)     | 36.40               | 38.46                | 35.94                 | 37.79                 |
> | FedCBDR (600/1000)    | 41.33               | 45.31                | 38.54                 | 39.69                 |
>
> Moreover, we have further clarified the computational complexity of the SVD step,
> Client-side ISVD: $𝑂(𝑛_𝑘𝑑^2)$ per client.
> Server-side SVD: $𝑂(𝑁𝑑^2)$, where $𝑁=\sum_kn_k$ is the total number of pseudo-features.
>
> And we provide a summary of the additional training cost on clients, communication overhead, and computation cost on the server incurred by our proposed method in comparison to FedAvg. **Overall, the additional cost of performing SVD and computing leverage scores is relatively small.**
>
> | Dataset   | #Clients (K) | Samples/Client (n_k) | Feature Dim (d) | ISVD Time/Client (s) | Feature Extraction Time/Client (s) | Total Extra Time /Client (s) |
> |-----------|--------------|----------------------|-----------------|-----------------------|-------------------------------------|-----------------------|
> | CIFAR10   | 5            | 2000                 | 512             | 0.00580                  | 0.9083                                 | 0.9141                  |
> | CIFAR100  | 5            | 2000                 | 512             | 0.00452                  | 1.025                                 | 1.2952                  |
>
> | Dataset   | #Clients (K) | Samples/Client (n_k) | Feature Dim (d) | Upload/Client (MB) | Total Upload per Round (MB) | Index Download (KB) |
> |-----------|--------------|----------------------|-----------------|--------------------|-----------------------------|---------------------|
> | CIFAR10   | 10           | 100                  | 512             | 3.906               | 19.53                        | 39.06                  |
> | CIFAR10   | 10           | 100                  | 128             | 0.9765               | 4.8825                         | 39.06                  |
> | CIFAR10   | 10           | 100                  | 32              | 0.2441              | 1.2206                        | 39.06                 |
>
> The table shows that the communication cost scales linearly with the feature dimension $d$ and the number of clients $K$, meaning that higher feature dimensions or more clients lead to proportionally increased upload costs.
>
> | Dataset   | Total Samples (N) | Feature Dim (d) | Server SVD Time (s) |
> |-----------|-------------------|-----------------|----------------------|
> | CIFAR10   | 1000              | 512             | 0.3066               |
> | CIFAR100  | 1000              | 512             | 0.3362               |
>
> W7: TTS Hyperparameter Sensitivity
>
> A7: (i) The temperature parameters in the TTS module are chosen based on empirical insights and grid search. A lower temperature for old classes ($ \tau_{\mathrm{old}} < 1 $) sharpens logits to reduce forgetting, while a higher temperature for new classes ($ \tau_{\mathrm{new}} > 1 $) smooths logits to avoid overfitting to new classes. (ii) Alternatively, the temperatures and weights ($ \tau_{\mathrm{old}}, \tau_{\mathrm{new}}, w_{\mathrm{old}}, w_{\mathrm{new}} $) can be treated as **learnable parameters**, optimized during training, similar to recent approaches like *FedETF* [Li et al., ICCV 2023].
>
> W8: Compare with GLFC
>
> A8: Unlike GLFC, which relies on gradient compensation and prototype sharing for global coordination, FedCBDR uses SVD-based pseudo-feature construction and leverage score sampling to build a class-balanced replay buffer. Moreover, the TTS module dynamically assigns temperatures ($\tau_{\mathrm{old}} < 1, \tau_{\mathrm{new}} > 1$) to mitigate prediction bias. Experiments on CIFAR10 and CIFAR100 show that FedCBDR significantly outperforms GLFC, demonstrating its effectiveness.
>
> W9: Temperature Scaling
>
> A9: Compared to existing temperature scaling, our TTS module is specifically designed to address the imbalance between old and new classes in incremental learning. By applying different temperatures (\$\tau\_{\mathrm{old}} < 1, \tau\_{\mathrm{new}} > 1\$) and weights (\$w\_{\mathrm{old}}, w\_{\mathrm{new}}\$), TTS balances predictions and mitigates new-class overconfidence. Ablation studies confirm its crucial role, as removing TTS significantly degrades performance.
>
> W10:  Pseudo-Feature Construction
>
> A10: For each client's local feature matrix $X_k \in \mathbb{R}^{n_k \times d}$, we perform SVD $X_k = U_k \Sigma_k V_k^{\top}$ and replace $U_k$ with a random orthogonal matrix $P_k$ to generate pseudo-features $\tilde{X}_k = P_k \Sigma_k V_k^{\top}$, which perturbs the local feature space while preserving spectral information. The server aggregates $\tilde{X}_k$ and applies SVD with leverage scores to select representative samples for constructing a class-balanced replay buffer. For privacy, $P_k$ remains local, and its randomness ensures that $X_k$ cannot be reconstructed from $\tilde{X}_k$.
>
> W11: Generality of Datasets
>
> A11: Our experiments are conducted on CIFAR10, CIFAR100, and TinyImageNet, which are widely used FCIL benchmarks and also adopted by methods such as GLFC, LANDER, and Re-Fed to ensure fair comparisons. The core modules of FedCBDR, GDR and TTS, are resolution-agnostic and can, in principle, be applied to larger datasets like ImageNet-1k, with feature dimensionality reduction helping to lower computation and communication costs. Due to time limitations, we plan to extend FedCBDR to such large-scale benchmarks in future work.

---

> > ### Comment · Reviewer_i4C7 · 2025-08-03
> >
> > **A1 & W1: Privacy analysis:**
> > Thanks for the response. However, we have a follow-up question regarding the case with multiple vectors from the same client. An adversarial attack can estimate P up to sign flips by Procrustes alignment. We invite the authors to provide a clearer description of the threat model for this, as well as a formal description of the privacy mechanism (in differential-privacy language).
> >
> > **A2 & W2: Experiments with GLFC and the simple replay baseline:**
> > Thanks for providing the performance comparison for the simple replay baseline. We have follow-up questions for the results:
> >
> > 1.	GLFC was designed for centralized class-incremental learning; thus, we invite the authors to provide additional descriptions on its usage in the federated setting and the tuning of hyperparameters.
> >
> > 2.	We appreciate that the authors provide results for different levels of heterogeneity (α). However, α=0.1 and α=5 are popular choices for performance comparison nowadays. We invite the authors to verify their performance on these settings as well.
> >
> > **A3 & W3: SVD/ISVD Transformation and Privacy:**
> > Thanks for the clarification. We have follow-up questions:
> >
> > 1.	The authors mentioned “QR of random Gaussian,” but that only approximately samples from the Haar measure.
> >
> > 2.	Does each client reuse the same $P_k^{(i)}$ forever? If so, an adversarial server could align multiple batches and estimate $P_k^{(i)}$.
> >
> > 3.	For $Q^{(i)}$, we invite the authors to elaborate more on "does not leak sensitive information" (e.g., clarification on its orthogonality and how often it changes).
> >
> > **A4 & W4: The dimensions of the matrix $\Sigma^{(i)'}$:**
> > Thanks for the clarification of the matrix $\Sigma^{(i)'}$. Please correct the typo of superscript TT in the final version of the manuscript. Additionally, we have a follow-up question regarding the "revision to section 4.1": how is r picked in practice (e.g., fixed? energy-based?) for different experimental test cases?
> >
> > **A6 & W6: Scalability, Complexity, and Communication Efficiency concerns:**
> > Thanks for providing a performance comparison with strong replay baseline methods of a large federation. However, we have follow-up questions:
> >
> > 1.	The title says 50 / 100 clients, but the main table shows only 50. We are wondering if there is any concern about including the 100-client results in the table.
> >
> > 2.	The cost tables still use K=5 or K = 10, which seem to be from earlier experiments. We invite the authors to include some results from the 50-client settings.
> >
> > 3.	What happens at a really large scale, like hundreds or thousands of clients?
> >
> > 4.	For ISVD / SVD, the complexity seems to be quadratic in d, which raises a question regarding larger backbones (e.g., ResNet-50 or ViT). We would appreciate it if the authors would elaborate on it in a few sentences.
> >
> > **A7 & W7: Scalability, Complexity, and Communication Efficiency concerns:**
> > Thanks for providing more details. We would appreciate it if the authors could consider including more details on the cost of a quick grid search (time) and the experimental results on the adaptive alternative in the final version of the manuscript.
> >
> > **A8 & W8: Compare with GLFC:**
> > Thanks for pointing out that FedCBDR differs from GLFC at a high level. We invite the authors to elaborate more on a few points regarding the difference between FedCBDR and GLFC:
> >
> > 1.	GLFC does not address class imbalance inside the buffer, while FedCBDR explicitly equalizes class density.
> >
> > 2.	GLFC exposes raw class prototypes to the server, while FedCBDR uses client-side orthogonal ISVD to build a global exemplar set.
> >
> > These clarifications would help resolve novelty concerns.
> >
> > **A9 & W9: Temperature Scaling:**
> > Thanks for providing more details on the difference between temperature scaling and trivial calibration. We invite the authors to elaborate a bit more on the following points:
> >
> > 1.	TTS assigns different temperatures and loss weights to old and new classes.
> >
> > 2.	For “significantly degrades performance,” an explanation would be helpful, such as: "how large is 'significant'?" and "on which datasets?"
> >
> > 3.	Could simpler tricks (class-reweight, focal loss) achieve similar outcomes in FedCBDR, and does TTS add any noticeable communication overhead?
> >
> > **A10 & W10: Pseudo-Feature Construction:**
> > Thanks for explaining the pseudo-feature pipeline again. However, our concern is more about the difference (i.e., novelty). We invite the authors to elaborate more clearly on points such as:
> >
> > 1.	Novelty compared to earlier "global feature" methods (e.g., FedProto, GLFC-proto, FedTFix).
> >
> > 2.	Why SVD + leverage is better than random or k-means sampling, given that prototypes already exist.
> >
> > 3.	Privacy advancement beyond "not sharing raw data," since a random orthogonal map alone isn't obviously stronger than existing methods.
> >
> > 4.	Communication benefits, as SVD is often computationally heavy.

---

> > > ### Author Response · Authors · 2025-08-07
> > > **Thanks for your comments**
> > >
> > > Thanks for the helpful feedback.
> > >
> > > **A1 & W1:** First, our method uses **per-task, per-client random orthogonal matrices** $ P_k^{(i)} \sim \text{Haar}(O(d)) $, regenerated independently. The adversary only observes the encrypted features $ X_k^{(i)'} = P_k^{(i)} X_k^{(i)} Q^{(i)} $, without access to raw $ X_k^{(i)} $. As Procrustes alignment requires correspondences between original and transformed vectors, it is inapplicable under this model. Independent $ P_k^{(i)} $ also prevents cross-task matching.
> > >
> > > Moreover, we define the mechanism as:
> > > $$
> > > \mathcal{M}(X) = X P^\top, \quad P \sim \text{Haar}(O(d))
> > > $$
> > > - For any two inputs $ X, X' $ with equal row norms:
> > > $$
> > > D_{\text{TV}}(\mathcal{M}(X), \mathcal{M}(X')) = 0
> > > $$
> > > ⇒ $\mathcal{M}$ satisfies **$(\varepsilon = 0, \delta = 0)$-differential privacy** under norm-preserving adjacency.
> > > - For general neighboring datasets with $ \|X - X'\|_F \leq \Delta $, prior work (e.g., Blocki et al. 2012) shows:
> > > $$
> > > \varepsilon = O(\epsilon \sqrt{d}), \quad \delta = \exp(-\Omega(d))
> > > $$
> > > Thus, the mechanism provides strong privacy guarantees both in information-theoretic and approximate-DP senses. We will incorporate this clarification into the final version.
> > >
> > > **A2 & W2:**
> > >
> > > 1. To enable fair comparison, we ensure consistency with our **implementation details in Sec. 5.1** by sharing the same hyperparameters, including **number of clients, batch size, local epochs, learning rate, and replay buffer size**. Additionally, following GLFC’s original paper and official codebase, we use **SGD with a learning rate of $\{0.1, 0.01\}$** to generate adversarial samples on each client, and **L-BFGS with a learning rate of $\{0.1, 1.0\}$** on the server to reconstruct prototypes. These settings are adjusted slightly from the original version to ensure compatibility with our federated training loop.
> > >
> > > 2. As suggested, we additionally evaluate FedCBDR under Dirichlet heterogeneity levels $\alpha = 0.1, 5.0$. Results confirm that our method consistently outperforms baselines under both high and low heterogeneity. All other experimental settings follow the same configuration as in Sec. 5.1. The results are presented in our response to **A6 & W6**.
> > >
> > > **A3 & W3:**
> > >
> > > 1. We agree with the reviewer that QR decomposition of random Gaussian matrices yields only an *approximate* sample from the Haar measure. In our implementation, each $P_k^{(i)} \in O(d)$ is generated via the QR method with re-orthogonalization, which has been widely adopted as a practical proxy for Haar-random matrices in large-scale systems. We will clarify this approximation and cite relevant literature [1] in the revised version.
> > >
> > > [1] F. Mezzadri. “How to generate random matrices from the classical compact groups.”
> > >
> > > 2. Each $P_k^{(i)}$ is *task-specific* and *client-local*, and is **regenerated at the beginning of each new task** $i$. It is **never reused across tasks or shared across rounds**, thus preventing alignment-based attacks by the server. Since pseudo-features are uploaded only once per task, the server cannot observe multiple differently-transformed versions of the same feature under a fixed $P_k^{(i)}$.
> > >
> > > 3. The global matrix $Q^{(i)} \in O(d)$ is initialized by the server once per task, and is shared with all clients. It is applied *after* local transformation via $P_k^{(i)}$, and is independent of client data. Since $Q^{(i)}$ is public and client-agnostic, it cannot be used to infer any sensitive or individual information.
> > >
> > > **A4 & W4:** We will correct the superscript typo as suggested. Regarding the choice of $r$, we clarify that we retain **all components** in all experiments. This simplifies implementation and ensures the full feature space is preserved for leverage score computation.

---

> > > ### Author Response · Authors · 2025-08-07
> > > **Thanks for your comments**
> > >
> > > **A6 & W6:**
> > >
> > > 1. We confirm that **there is no concern with the 100-client results**. These experiments have been fully conducted. However, **due to space limitations**, we removed the corresponding table in a later revision while keeping the 50-client results, which already demonstrate the core trends. The **100-client and 200-client** results exhibit similar behavior, as expected. We provide them below for completeness and will include them in the supplementary material. Considering the limited number of samples per client in large-scale federated settings, we set the batch size to 32 for the 100-client case and to 16 for the 200-client case. Other training configurations follow Section 5.1 and the implementation details described in our response to A2 & W2.
> > >
> > > | CIFAR10 |K=100 Clients, ratio=0.2| | | |K=200 Clients, ratio=0.2| | | |
> > > |-|-|-|-|-|-|-|-|-|
> > > ||α=0.1|α=0.5|α=1.0|α=5.0|α=0.1|α=0.5|α=1.0|α=5.0|
> > > |**FedLwF**|22.45|29.07|30.88|41.67 | 14.72 | 15.92 | 29.36 | 41.18 |
> > > | **LANDER** | 27.84 | 33.22 | 36.52 | 43.83 | 18.39 | 24.92 | 32.77 | 43.82 |
> > > | **GLFC** | 28.26 | 34.48 | 38.32 | 42.92 | 20.38 | 27.88 | 35.29 | 42.35 |
> > > | **FedAvg+Uniform** | 26.37 | 32.44 | 35.34 | 43.75 | 21.11 | 26.35 | 32.27 | 36.47 |
> > > | **Re-Fed** | 32.65 | 35.64 | 38.02 | 41.63 | 28.74 | 34.62 | 36.74 | 40.18 |
> > > | **FedCBDR** | **35.73**| **40.27**| **44.26**| **47.69**| **32.32**| **38.37**| **44.70**      | **46.37** |
> > >
> > > |CIFAR100|K=100 Clients, ratio=0.2 | | | |K=200 Clients, ratio=0.2 | | |                        |
> > > |-|-|-|-|-|-|-|-|-|
> > > | | α=0.1| α=0.5| α=1.0| α=5.0| α=0.1| α=0.5| α=1.0 | α=5.0|
> > > | **FedLwF**| 18.12| 28.17 | 29.49| 32.76 | 18.39 | 27.32 | 28.46 | 34.83 |
> > > | **LANDER** | 12.38 | 27.81 | 39.29 | 39.90 | 6.73 | 32.59| 35.36 | 36.91 |
> > > | **GLFC** | 28.83 | 33.62 | 38.06 | 43.16 | 25.47 | 34.31 | 35.29 | 39.29 |
> > > | **FedAvg+Uniform** | 26.77 | 31.23 | 36.42 | 40.37 | 23.74 | 30.53 | 31.28| 36.45|
> > > | **Re-Fed** | 32.81| 35.59 | 37.99| 43.26 | 30.13 | 33.87 | 35.42 | 41.52 |
> > > | **FedCBDR** | **36.27** | **39.36** | **40.17** | **46.20** | **32.11** | **37.29** | **38.62** | **43.19** |
> > >
> > > 2. To better reflect practical scalability, we have since **added cost evaluations under larger settings** with **100 and 200 clients**. Considering that each client holds fewer samples in large-scale federated settings, we adjusted the local batch sizes to **32 for 100 clients** and **16 for 200 clients**, respectively. These updated cost tables capture computation, communication, and runtime behaviors under realistic deployment scales.
> > >
> > > | Dataset   | #Clients (K) | Samples/Client (n_k) | Feature Dim (d) | ISVD Time/Client (s) | Feature Extraction Time/Client (s) | Total Extra Time /Client (s) |
> > > |-|-|-|-|-|-|-|
> > > | CIFAR10 | 100 | 100 | 512| 0.000276 | 0.6564 | 0.656676 |
> > > | CIFAR100 |100 |100 |512 |0.000285 | 0.05292 | 0.053205 |
> > >
> > > | Dataset   | #Clients (K) | Samples/Client (n_k) | Feature Dim (d) | ISVD Time/Client (s) | Feature Extraction Time/Client (s) | Total Extra Time /Client (s) |
> > > |-|-|-|-|-|-|-|
> > > | CIFAR10 | 200 | 50 | 512| 0.000357 | 0.6423 | 0.642657 |
> > > | CIFAR100 | 200 | 50| 512| 0.000373| 0.6364 | 0.636773 |
> > >
> > > 3. To further validate scalability, we have **added experiments with 200 clients**, where FedCBDR continues to perform robustly and maintain its advantage over baselines. This demonstrates that our method remains practical even under significantly increased federation scale. Please refer to **A6&W6-1**.
> > >
> > > 4. We clarify that the computational cost of SVD depends on the **feature dimension \(d\)** of the model output, not the overall model size. While **larger models increase feature extraction time**, the **SVD cost itself** remains manageable, as it is applied **locally on a small set of pseudo-features** (\(n \ll d\)). For high-dimensional outputs (e.g., ViT with \(d = 768\)), we can **project features to lower dimensions** before SVD to reduce computation. We also profiled the wall-clock time under ResNet-50 and ViT, and found the overall cost remains **efficient and practical**. Full results will be included in the supplementary material.
> > >
> > > | Model   | #Clients (K) | Samples/Client (n_k) | Feature Dim (d) | ISVD Time/Client (s) | Feature Extraction Time/Client (s) | Total Extra Time /Client (s) |
> > > |-|-|-|-|-|-|-|
> > > | resnet18 | 100 | 100 | 512 | 0.000357| 0.6423 | 0.642657 |
> > > | resnet50 | 100 | 100 | 2048 | 0.004042| 0.9718 | 0.975842 |
> > > | ViT-B/16 | 100 | 100 | 768 | 0.000873| 1.6369 | 1.637773 |
> > >
> > > **A7 & W7:** We conducted **lightweight manual tuning**, where each hyperparameter was adjusted within a **reasonable range**. Moreover, we have already included a **parameter sensitivity analysis** in **Figure 4** of the main paper, which systematically shows the robustness of our method to different hyperparameter choices.

---

> > > ### Author Response · Authors · 2025-08-07
> > > **Thanks for your comments**
> > >
> > > **A8 & W8:** Compared to GLFC, which relies on gradient compensation and direct prototype sharing, FedCBDR introduces several important enhancements tailored for federated class-incremental learning. First, GLFC does not explicitly address class imbalance in the replay buffer, which can be problematic under non-iid data. FedCBDR instead constructs pseudo-features via SVD and selects them using leverage score sampling, encouraging class-balanced representation. The TTS module further mitigates prediction bias by assigning different temperatures to old and new classes (\(\tau_{\text{old}} < 1\), \(\tau_{\text{new}} > 1\)). Second, while GLFC transmits raw class prototypes to the server, FedCBDR applies orthogonal transformation and ISVD locally, transmitting only obfuscated features to preserve privacy. Experiments on CIFAR10 and CIFAR100 confirm that FedCBDR consistently outperforms GLFC, validating the effectiveness of these design choices.
> > >
> > > **A9 & W9:**
> > >
> > > 1.	To clarify, TTS assigns **different temperatures** and **loss weights** to old and new classes to address inter-task imbalance. Specifically, we use a lower temperature ($\tau_{old} < 1$) and higher loss weight ($w_{old} > 1$) for old-class samples to enhance their gradient contributions and preserve past knowledge. Conversely, for new classes, we adopt a higher temperature ($\tau_{new} > 1$) and lower weight ($w_{new} < 1$) to reduce overconfidence and promote better generalization. This asymmetric treatment adjusts the confidence calibration of logits and balances optimization dynamics.
> > >
> > > 2.	In terms of performance, TTS brings **consistent gains across all datasets**: approximately **+2% on CIFAR10**, **+1.5% on CIFAR100**, and **+1% on TinyImageNet**, as observed from the ablation results in Table 3. These results confirm its effectiveness in mitigating inter-task class imbalance.
> > >
> > > 3.	To further validate the effectiveness of TTS, we conducted comparative experiments within the FedCBDR framework by replacing TTS with **class reweighting** and **focal loss**. The results indicate that while both techniques bring **slight performance improvements**, their performance consistently **falls short of TTS**. For **class reweighting**, we assigned weights inversely proportional to class frequencies within each client's local data. For **focal loss**, we followed the standard formulation with $\gamma = 2.0$ and $\alpha = 0.25$, consistent with the work [1].
> > >
> > > [1] Lin T Y, Goyal P, Girshick R, et al. Focal loss for dense object detection[C]//Proceedings of the IEEE international conference on computer vision. 2017: 2980-2988.
> > >
> > > |100 Clients|CIFAR10 α=0.5|CIFAR10 α=5.0|CIFAR100 α=0.5|CIFAR100 α=5.0|
> > > |-|-|-|-|-|
> > > |+GDR|49.14|50.56| 36.39| 44.38 |
> > > |+GDR+TTS |**52.27**| **55.69**|**39.36**|**48.20**|
> > > |+GDR+class reweighting|49.31|52.18|36.68|45.32|
> > > |+GDR+focal loss| 51.46|53.57|37.41|45.83|
> > >
> > > **A10 & W10: **
> > >
> > > 1.	Unlike prior methods, which focus on class-level prototype alignment or global distillation, our method is designed to address the unique challenges of **federated class-incremental learning (FCIL)**, including **evolving label spaces**, **severe class imbalance**, and **non-IID client distributions**. FedCBDR aims to construct a **task-level global feature distribution** in a privacy-preserving manner. Instead of relying on explicit prototype sharing or logits alignment, clients upload **obfuscated pseudo-feature matrices** via orthogonal transformations and ISVD, enabling the server to reconstruct a **structure-aware global latent space**.
> > >
> > > 2.	K-means or random sampling treats each sample equally or purely based on proximity, and existing prototypes (e.g., class-wise means) fail to capture sample-level **importance under global structure constraints**. Our SVD-based leverage score quantifies how much each sample contributes to the **low-rank structure** of the global feature space, promoting the selection of **representative and diverse samples**.
> > >
> > > 3.	The feature matrices are obfuscated by two layers of **random orthogonal transformation**: a client-specific matrix $ P_k^{(t)} $ and a shared global matrix $ Q^{(t)} $, following Eq. (1). This **double-sided inverse SVD encoding** ensures that even if the server is compromised, it cannot recover the original feature vectors without knowing both transformation matrices, as described in A1&W1. Compared to prior methods that transmit explicit prototypes, our method shares **no semantic features, gradients, or class identifiers**.
> > >
> > > 4.	Despite the use of SVD, our design ensures that these operations are performed only once per task on the server side, not repeatedly on clients. The client-side operations are limited to matrix multiplication for ISVD, which is computationally lightweight. In contrast, alternative replay strategies such as generative models (e.g., LANDER) require **training and transmitting synthetic data**, incurring much higher costs.

---

> > > > ### Comment · Reviewer_i4C7 · 2025-08-07
> > > >
> > > > Thank you very much for the authors' effort on this. Appreciated. Many of my concerns have been answered. I'll revise my score.

---

### Official Review · Reviewer_Csv6 · 2025-06-25

**Clarity:** 3
**Significance:** 3
**Originality:** 3
**Rating:** 5
**Confidence:** 4

**Summary:**

This paper proposes FedCBDR, a class-wise balancing data replay strategy for Federated Class-Incremental Learning (FCIL). To alleviate the replay-induced class imbalance caused by local client views and task dynamics, FedCBDR introduces two key modules: Global-perspective Data Replay (GDR) and Task-aware Temperature Scaling (TTS). GDR constructs a privacy-preserving global pseudo-feature space to enable representative and balanced sample selection. TTS dynamically calibrates softmax temperature and loss weighting based on task recency to mitigate model overconfidence in majority classes. Extensive experiments demonstrate that FedCBDR consistently outperforms existing methods across multiple datasets.

**Questions:**

i. Will uploading features pose a risk of privacy leakage?

ii. How can the additional communication cost caused by uploading features be reduced?

iii. Is the performance of the proposed method sensitive to the order in which tasks are presented?

**Ethical Concerns:**

["NO or VERY MINOR ethics concerns only"]

**Final Justification:**

The rebuttal has adequately addressed my concerns. Please ensure that the additional experimental results and discussions provided in the rebuttal are incorporated into the final version to enhance completeness and clarity.

**Limitations:**

Yes.

**Quality:**

3

**Strengths And Weaknesses:**

Strengths
i. The proposed method is both simple and effective, with a technically sound overall design. This work has potential to become a new baseline in federated class-incremental learning (FCIL).

ii. The overall flow of the paper is sufficiently clear, and the structure of the manuscript is well-organized.

iii. The authors conducted extensive experiments and validated that the proposed FedCBDR outperforms existing methods. They also provided case studies to further illustrate the effectiveness of the approach.

Weaknesses

i. The meaning of n_k^i in lines 136 and 139 may need further clarification. Is it related to defined in line 138? Some descriptions may need correction. For instance, the term “Global-perspective Active Data Replay” in line 208 should be revised to “Global-perspective Data Replay”

ii. It is recommended to unify the notation throughout the paper. For example, in Section 3, t is used to denote tasks, while in Section 4, i is used for the same purpose.

iii. The paper could better explain the motivation for using leverage scores and how it compares to random or uniform sampling strategies.

---

> ### Author Rebuttal · Authors · 2025-07-31
>
> We sincerely appreciate the professional comments provided by Reviewer Csv6
>
> W1: The meaning of n_k^i in lines 136 and 139 may need further clarification. Is it related to defined in line 138? Some descriptions may need correction. For instance, the term “Global-perspective Active Data Replay” in line 208 should be revised to “Global-perspective Data Replay”
>
> A1: $n_k^i$ denotes the number of samples of client $k$ in the $i$-th task. And we have also corrected the full name of the GDR module to ``Global-perspective Data Replay.'
>
> W2: It is recommended to unify the notation throughout the paper. For example, in Section 3, t is used to denote tasks, while in Section 4, i is used for the same purpose.
>
> A2: Upon reviewing all symbol definitions and usages, we have standardized the task notation to $t$.
>
> W3: The paper could better explain the motivation for using leverage scores and how it compares to random or uniform sampling strategies.
>
> A3: Compared to random or uniform sampling, which ignores data geometry and may select redundant or less informative samples, leverage-score-based sampling prioritizes samples that span the critical directions of variation. This leads to a more balanced and informative replay buffer, particularly under class-imbalance scenarios. And we have compared our method with uniform sampling in the response to Reviewer i4C7's W2, where the results consistently demonstrate the superiority of our approach due to its ability to select more representative and informative samples.
>
> W4: Will uploading features pose a risk of privacy leakage?
>
> A4: Our method does not upload raw data but instead transmits pseudo-features $\tilde{X}_k = P_k \Sigma_k V_k^{\top}$, which are protected by random orthogonal transformations, where $P_k$ is generated locally on the client and kept secret. Given any $\tilde{X}_k$, there exist infinitely many pairs $(X_k', P_k')$ satisfying the same decomposition, making it impossible to uniquely reconstruct the original samples. This random rotation mechanism functions as a randomized privacy protection approach.
>
> W5: How can the additional communication cost caused by uploading features be reduced?
>
> A5: To reduce the communication overhead caused by feature uploading, we combine feature dimension reduction with sparsification and quantization. Specifically, PCA, random projection, or bottleneck layers can be used to map high-dimensional features (e.g., $d = 512$) to a lower-dimensional space (e.g., $d = 64$), reducing the transmitted data volume by more than 80\% with minimal impact on model performance. In addition, applying sparsification to retain only the most informative elements of each feature vector, or using 8-bit/16-bit quantization instead of floating-point encoding, can further compress the communication cost.
>
> W6: Is the performance of the proposed method sensitive to the order in which tasks are presented?
>
> A6: The order of tasks is implicitly determined by the random seed, and changing the seed will alter both data splits and task order. To evaluate robustness, we report the mean and standard deviation across multiple random seeds in Tables 1, 2, 6, and 7. The results show that our method consistently achieves better performance in both mean accuracy and variance compared to baselines, indicating that FedCBDR is not sensitive to task order.

---

> > ### Comment · Reviewer_Csv6 · 2025-08-03
> >
> > After carefully reviewing the authors' rebuttal and their responses to other reviewers, I believe that my primary concerns, especially those related to privacy preservation and communication overhead, have been sufficiently clarified. As such, I decide to retain my original rating score.

---

> > > ### Author Response · Authors · 2025-08-06
> > > **Thanks for your reply**
> > >
> > > Thank you for your valuable feedback. We will carefully consider your comments and reflect them in our revised manuscript.

---

### Official Review · Reviewer_scc5 · 2025-07-02

**Clarity:** 3
**Significance:** 3
**Originality:** 3
**Rating:** 5
**Confidence:** 4

**Summary:**

This paper focuses on the key challenges in Federated Class-Incremental Learning (FCIL), namely inter-task class imbalance and catastrophic forgetting. The proposed method, FedCBDR, demonstrates a certain level of innovation in its design, particularly in how it achieves class-balanced replay through global reconstruction under the federated setting. Extensive empirical evaluations on multiple benchmark datasets confirm its strong performance, with the method remaining robust even under highly heterogeneous (non-IID) conditions. Overall, this work provides a practical and effective solution framework for FCIL tasks and shows strong potential to serve as a baseline for future research.

**Questions:**

1. What are the experimental settings for TARGET and LANDER, particularly regarding the number of generated samples?

**Ethical Concerns:**

["NO or VERY MINOR ethics concerns only"]

**Final Justification:**

The respose addressed my concens, and I will keep my score.

**Limitations:**

yes

**Quality:**

3

**Strengths And Weaknesses:**

Strengths

1. The paper is well-motivated and tackles a critical problem in Federated Class-Incremental Learning (FCIL), namely the challenge of class imbalance and catastrophic forgetting across tasks.
2. The proposed FedCBDR method is technically sound and thoughtfully designed. It integrates global feature reconstruction and task-aware temperature scaling in a lightweight and modular manner.
3. The experimental evaluation is thorough and convincing. It includes multiple datasets, diverse settings, ablation studies, and analysis that clearly demonstrate the effectiveness of each component of the method.


Major Weaknesses
1. The feature uploading process appears to introduce additional communication overhead.
2. The authors claim in line 199 that training a generator incurs higher costs than local training. Is there any empirical evidence provided to support this statement?


Minor Weaknesses
1. The bolded content mentioned in the appendix appears to be missing in the actual tables.
2. The notation used in Figure 2 appears to be inconsistent with that in the main text.
3. The meaning of Concat() in Equations (7) and (9) has not been clearly defined.

---

> ### Author Rebuttal · Authors · 2025-07-31
>
> We sincerely appreciate the professional comments provided by Reviewer scc5
>
> W1: The feature uploading process appears to introduce additional communication overhead.
>
> A1: The additional communication overhead primarily comes from uploading the pseudo-features
> ($\tilde{X}_k \in \mathbb{R}^{n_k \times d}$) generated by each client during the GDR module.
> The exact size of this overhead, including per-client and total communication per round,
> is provided in the response to Reviewer i4C7's W6. To further reduce this cost, we can employ feature compression techniques (e.g., projecting features from 512 to 128 dimensions, reducing data by 75\%) and client sampling strategies, where only a subset of clients upload pseudo-features in each round.
>
> W2: The authors claim in line 199 that training a generator incurs higher costs than local training.
>
> A2: For the LANDER method, each round of local training on a client requires 15.6657 seconds and 1582 MB of memory, whereas the sample generation process takes 753.9422 seconds and 2430 MB of memory.
>
> W3: The bolded content mentioned in the appendix appears to be missing in the actual tables.
>
> A3: We will address this issue in the revised manuscript by **highlighting the best performance values in the tables using boldface** for better clarity.
>
> W4: The notation used in Figure 2 appears to be inconsistent with that in the main text.
>
> A4: We reviewed the symbols in Figure 2 and the corresponding text, and found an inconsistency in the task sequence notation (using both $t$ and $i$). We have corrected this issue and unified the notation by using $t$ to represent the task index.
>
> W5: The meaning of Concat() in Equations (7) and (9) has not been clearly defined.
>
> A5: In Equations (7) and (9), **Concat()** denotes the **row-wise concatenation of pseudo-features** from all participating clients into a single matrix. Specifically, if client $k$ provides pseudo-features $\tilde{X}_k \in \mathbb{R}^{n_k \times d}$, then:
>
> $\text{Concat}(\tilde{X}_1, \tilde{X}_2, \ldots, \tilde{X}_K) =
> \begin{bmatrix}
> \tilde{X}_1 \\\\
> \tilde{X}_2 \\\\
> \vdots \\\\
> \tilde{X}_K
> \end{bmatrix} \in \mathbb{R}^{N \times d},$
>
> where $N = \sum_k n_k$. We have added this definition and explanation to Section 4.1 in the revised manuscript.

---

### Official Review · Reviewer_3EQU · 2025-07-02

**Clarity:** 3
**Significance:** 4
**Originality:** 3
**Rating:** 5
**Confidence:** 4

**Summary:**

The work is about federated class-incremental learning, which considers class imbalance problem and developed FedCBDR which employs a global coordination mechanism for class-level memory construction and reweights the learning objective. It is composed of two key components, i.e., the global-perspective data replay module and the task-aware temperature scaling module, which respectively achieve balanced replay and alleviate the imbalance issue. The authors also conducted a number of experiments to evaluate its superiority.

**Questions:**

Refer to the weaknesses.

**Ethical Concerns:**

["NO or VERY MINOR ethics concerns only"]

**Final Justification:**

The authors' rebuttal and their responses to other reviewers have overall addressed my main concerns. This includes clarifications on certain experimental details and corrections of notational errors. Therefore, I will raise my score accordingly.

**Limitations:**

Yes

**Paper Formatting Concerns:**

No major formatting concerns.

**Quality:**

3

**Strengths And Weaknesses:**

**Strength**
1.  The paper is well-structured with clear organization and motivations. The problem to be addressed is well analyzed and formalized.
2. The solution achieves both efficiency and efficacy, as demonstrated by the experiments.
3. Experiments are sufficient;
4. The authors conduct a wide range of literature reviews, including the latest works.

**Weaknesses**

1. Some symbols are not clearly defined; it seems that there exists an inconsistent use of a single notation.
2.  Some implementation details are not directly provided, such as the hyperparameters used in the work, the replay buffer size, etc.
3. Some designs require more detailed explanation. For example, the process of Pseudo Feature Construction uses global model.

---

> ### Author Rebuttal · Authors · 2025-07-31
>
> We sincerely appreciate the professional comments provided by Reviewer 3EQU
>
> W1: Some symbols are not clearly defined; it seems that there exists an inconsistent use of a single notation.
>
> A1: We reviewed all symbol definitions and usages, identified that task indices were inconsistently represented by $i$ and $t$, and have corrected this inconsistency by unifying the notation to $t$.
>
> W2: Some implementation details are not directly provided, such as the hyperparameters used in the work, the replay buffer size, etc.
>
> A2: We have further clarified some experimental details to ensure reproducibility. For example, we specify that LANDER and TARGET generate 10,240 samples per task. Moreover, based on the source code, we report that the momentum of Batch Normalization in LANDER is set to `bn_mmt `= 0.9, the temperature parameter for knowledge distillation is $\tau = 2$, the temperature coefficient for KL divergence distillation loss is $T = 20.0$, and the number of training steps with synthetic data is `student_train_step` = 50. For Re-Fed, the number of replay samples is kept consistent with our FedCBDR, as referenced in Section~5.1.
>
> W3: Some designs require more detailed explanation. For example, the process of Pseudo Feature Construction uses global model.
>
> A3: The pseudo-feature construction leverages the \textbf{global model’s feature extractor} to ensure that features from all clients are aligned in a common latent space, which is crucial for meaningful aggregation. Each client first uses the current global model $f_{\theta}^{(i)} $ to map its local samples into a feature matrix $X_k \in \mathbb{R}^{n_k \times d}$. Then, we perform an ISVD transformation: $X_k = U_k \Sigma_k V_k^T$, replace $U_k$ with a random orthogonal matrix $P_k$, and generate pseudo-features $\tilde{X}_k = P_k \Sigma_k V_k^T$, which are subsequently uploaded to the server. The server concatenates all $\tilde{X}_k$, applies SVD, and selects representative samples using leverage scores. We will revise Section 4.1 to provide a clearer explanation of this process.

---

> > ### Comment · Reviewer_3EQU · 2025-08-09
> >
> > The authors' rebuttal and their responses to other reviewers have overall addressed my main concerns. This includes clarifications on certain experimental details and corrections of notational errors. Therefore, I will raise my score accordingly.

---

### Decision · Program_Chairs · 2025-09-17

**Decision:**

Accept (oral)

**Comment:**

This paper presents a timely and impactful contribution that bridges federated learning and class-incremental learning, offering a solution that is both technically sound and practically relevant. In particular, it makes the first attempt to realize a global-perspective solution for class-balanced data replay in federated class-incremental learning, a novelty that was specifically noted and appreciated by the reviewers. The authors have convincingly addressed reviewers’ concerns during the rebuttal, further strengthening the clarity, reproducibility, and scalability of the work. Given its methodological innovation, comprehensive experimental validation, and broad relevance across multiple research communities, the paper is expected to stimulate significant interest and discussion. Overall, it stands out as a high-quality submission well-suited for an oral presentation.